# Physics-based SNOWPACK model improves representation of near-surface Antarctic snow and firn density

Eric Keenan[1], Nander Wever[1], Marissa Dattler[2], Jan T. M. Lenaerts[1], Brooke Medley[3], Peter Kuipers Munneke[4], and Carleen Reijmer[4]

[1]Department of Atmospheric and Oceanic Sciences, University of Colorado, Boulder, CO, USA
[2]Department of Atmospheric and Oceanic Sciences, University of Maryland, College Park, MD, USA
[3]Cryospheric Sciences Laboratory, NASA Goddard Space Flight Center, Greenbelt, MD, USA
[4]Institute for Marine and Atmospheric Research, Utrecht University, Utrecht, The Netherlands

**Correspondence:** Eric Keenan (eric.keenan@colorado.edu)

**Abstract.** Estimates of snow and firn density are required for satellite altimetry based retrievals of ice sheet mass balance that rely on volume to mass conversions. Therefore, biases and errors in presently used density models confound assessments of ice sheet mass balance, and by extension, ice sheet contribution to sea level rise. Despite this importance, most contemporary firn densification models rely on simplified semi-empirical methods, which are partially reflected by significant modeled density errors when compared to observations. In this study, we present a new drifting snow compaction scheme that we have implemented into SNOWPACK, a physics-based land surface snow model. We show that our new scheme improves over existing versions of SNOWPACK by increasing simulated near-surface (defined as the top 10 m) density to be more in line with observations (near-surface bias reduction from -44.9 to -5.4 $\mathrm{kg\,m^{-3}}$). Furthermore, we demonstrate high-quality simulation of near-surface Antarctic snow and firn density at 122 observed density profiles across the Antarctic ice sheet, as indicated by reduced model biases throughout most of the near-surface firn column when compared to two semi-empirical firn densification models (SNOWPACK mean bias = -9.7 $\mathrm{kg\,m^{-3}}$, IMAU-FDM mean bias = -32.5 $\mathrm{kg\,m^{-3}}$, GSFC-FDM mean bias = 15.5 $\mathrm{kg\,m^{-3}}$). Notably, our analysis is restricted to the near-surface where firn density is most variable due to accumulation and compaction variability driven by synoptic weather and seasonal climate variability. Additionally the GSFC-FDM exhibits lower mean density bias from 7 - 10 m (SNOWPACK bias = -22.5 $\mathrm{kg\,m^{-3}}$, GSFC-FDM bias = 10.6 $\mathrm{kg\,m^{-3}}$) and throughout the entire near-surface at high accumulation sites (SNOWPACK bias = -31.4 $\mathrm{kg\,m^{-3}}$, GSFC-FDM bias = -4.7 $\mathrm{kg\,m^{-3}}$). However, we found that the performance of SNOWPACK did not degrade when applied to sites that were not included in the calibration of semi-empirical models. This suggests that SNOWPACK may possibly better represent firn properties in locations without extensive observations and under future climate scenarios, when firn properties are expected to diverge from their present state.

## 1 Introduction

The Antarctic ice sheet (AIS) is the largest freshwater reservoir on Earth, and if melted entirely would raise globally averaged sea level by 58 m (The IMBIE team, 2018). The AIS is contributing to sea level rise via net mass loss, at an increasing rate from $40 \pm 9$ Gt yr$^{-1}$ in 1979 - 1990 to $252 \pm 26$ Gt yr$^{-1}$ in 2009 - 2017 (Rignot et al., 2019). In order to quantify ice sheet

contribution to sea level rise, glaciologists compute the mass balance (MB), defined as the difference between the grounded ice sheet surface mass balance (SMB) and solid ice discharge across the grounding line (Lenaerts et al., 2019). MB is typically calculated using one of three methods, namely the input output method (Rignot et al., 2019), gravimetry (Velicogna et al., 2020), or satellite altimetry (e.g. Shepherd et al., 2012; Smith et al., 2020), the latter of which combines measurements of ice sheet volume change with modeled snow and firn density estimates. Similar to MB, spatial variations in SMB can be determined from combined density and radar derived annual snow accumulation estimates (e.g. Medley et al., 2013; Dattler et al., 2019; Kausch et al., 2020). However, in all cases, density estimates represent one of the largest sources of uncertainty (Shepherd et al., 2012; Montgomery et al., 2020) due to an uncertain and simplified representation of snow and firn densification (Alexander et al., 2019; Montgomery et al., 2020), particularly over the low-accumulation interior of ice sheets (Weinhart et al., 2020).

Antarctic new snow density and subsequent densification are known to vary in both space and time and are influenced by, among other factors, local meteorology and drifting snow (Herron and Langway, 1980; Sommer et al., 2018). In particular, surface snow and firn density are known to be strongly impacted by wind-driven compaction, a process hereafter referred to as drifting snow compaction, whereby mobilized snow particles in the saltation layer (within the lowermost 2 m of the atmosphere) break apart upon collision with the snow surface. This process results in fragmented and rounded grains which pack together more efficiently, resulting in increased density (Vionnet et al., 2012). Drifting snow occurs up to 75 % of the time in the AIS escarpment zone (Lenaerts et al., 2012; Palm et al., 2017; van Wessem et al., 2018; Amory and Kittel, 2019), and because of drifting snow redistribution, precipitating snow particles are not always permanently incorporated into the snowpack at the time or location of precipitation. In fact, observations show that in the high-elevation, dry, and windy areas of the AIS, pockets of fresh snow can be found alongside snow that is more than one year old (Picard et al., 2019). Including processes such as drifting snow compaction, snow metamorphism, and compaction due to overburden stress has been shown to improve model representation of polar snow and firn (van Kampenhout et al., 2017). However, despite this finding, as well as the demonstrated complexity of snow and firn densification, many AIS MB studies (e.g. Zwally et al., 2015; Smith et al., 2020) rely on relatively simplified semi-empirical models (e.g. Ligtenberg et al., 2011; Li and Zwally, 2011; Medley et al., 2020). Semi-empirical densification models successfully capture broad regional variability in firn characteristics (van den Broeke, 2008; Ligtenberg et al., 2011). However, due to their limited complexity, as measured by the inclusion of ephemeral processes such as drifting snow compaction, they cannot capture high frequency variability in near surface snow density originating from varying atmospheric conditions such as temperature and wind speed. Such variability, which can act on hourly time scales, is known to exist from field observations (e.g., Sommer et al., 2018), and may therefore drive erroneous density estimates that are ultimately used in satellite altimetry volume to mass conversions.

Additionally, semi-empirical models are tuned to observations representative of the past or present climate (Herron and Langway, 1980; Ligtenberg et al., 2011; Li and Zwally, 2011), which may not be representative under future climate change scenarios (Ligtenberg et al., 2014). Furthermore, because semi-empirical models rely on extensive tuning to observations, they may not perform well in climates poorly sampled by observations (e.g. the East Antarctic plateau). Alternatively, physics based models, for example CROCUS (Vionnet et al., 2012) and SNOWPACK (Bartelt and Lehning, 2002; Lehning et al., 2002a, b), do not explicitly tune simulated density to observations. Instead, physics-based models represent densification using

a constitutive relationship between stress and strain for snow. However, physics-based approaches are hindered by a variety of factors including computational expense, the requirement to estimate unknown parameters such as surface roughness length (controlling turbulent fluxes and friction velocity) and snow activation energy (controlling snow viscocity), as well as the need to calculate intermediate prognostic variables including thermal conductivity, viscosity, and snow grain shape and size. Despite these drawbacks, the complex material properties of snow, combined with an acute scarcity of snow and firn density measurements lead us to hypothesize that a physics-based modeling approach is preferred.

In order to improve model representation of Antarctic snow and firn properties compared to semi-empirical models, we compare results from the physics-based snow model, SNOWPACK, forced by hourly weather data from MERRA-2 atmospheric reanalysis (section 2.2) to nine automatic weather stations (AWSs), 55 borehole 10 m depth temperatures, and 122 observed near-surface density profiles for a total of 186 locations across the AIS. First, we describe model setup and a new drifting snow routine designed to improve representation of near-surface (depth $\leq 10$ m) snow and firn density (section 2.1). Next, we evaluate SNOWPACK's ability to simulate the surface energy balance by comparing with available surface temperature proxies (section 3.1). We then explore the sensitivity of SNOWPACK simulated density profiles to uncertainties in atmospheric forcing and prescribed snow physics (sections 3.3 - 3.4). Next, we compare SNOWPACK to observed density profiles and compare the relative performance of SNOWPACK to two other firn densification models (sections 3.5 - 3.8). Finally, we conclude with a discussion of SNOWPACK predicted surface density variability and its implications for satellite altimetry based measurements of ice sheet MB (section 3.9).

## 2 Methods

### 2.1 Physics-based snow model SNOWPACK

Models commonly assume a high ($> 250 \mathrm{~kg~m^{-3}}$) new snow density over Antarctica (Ligtenberg et al., 2011; Groot Zwaaftink et al., 2013) despite observational evidence of occasional relatively low ($< 100 \mathrm{~kg~m^{-3}}$) new snow density (Groot Zwaaftink et al., 2013; Sommer et al., 2018). This difference can be explained by accumulation of snowfall without sufficient wind speed required for drifting snow compaction to occur. In order to account for this, we have implemented a new drifting snow compaction routine into the vertical, one-dimensional physics-based land-surface snow model, SNOWPACK (Bartelt and Lehning, 2002; Lehning et al., 2002a, b), which contrasts to most existing firn models (sections 2.5 - 2.6) in that it calculates densification using an overburden stress formulation as opposed to an empirical relationship and explicitly determines snow viscosity by calculating the snow microstructure (e.g. grain size and shape) and temperature. Originally designed as an alpine snow cover model capable of describing snow cover properties, including accumulation, densification, temperature, energy balance, and snow microstructure, SNOWPACK has been applied to both the Greenland and Antarctic ice sheets (Groot Zwaaftink et al., 2013; Van Tricht et al., 2016; Steger et al., 2017; Izeboud et al., 2020; Dunmire et al., 2020). These studies have shown that SNOWPACK is capable of capturing important processes in the ice sheet firn layer, namely accumulation in windy environments, surface meltwater production, and subsequent liquid water retention in the firn. However, these studies have not implemented a polar snow accumulation scheme that allows for initial accumulation of low ($< 250 \mathrm{~kg~m^{-3}}$) density snow and

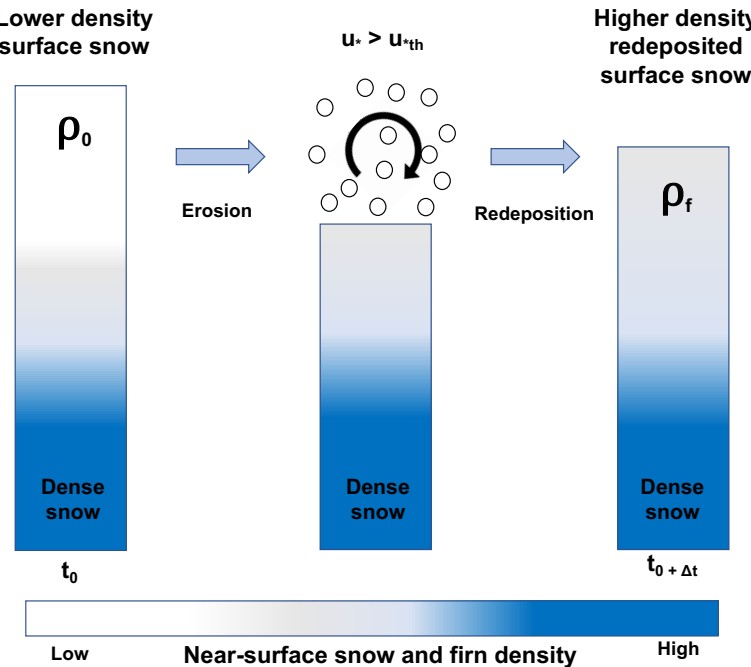

**Figure 1. Schematic of SNOWPACK drifting snow compaction.** When the friction velocity exceeds the snow threshold friction velocity ($u_* > u_{*\text{th}}$), initial lower density surface snow (left) is eroded by the wind, suspended above the snow surface (middle), and then redeposited with a higher density (right).

subsequent drifting snow compaction. Additionally, no previous study has evaluated SNOWPACK simulated near-surface firn density across the entire AIS.

Our new drifting snow compaction routine combines an existing alpine new snow density parameterization with a polar snow compaction routine. In particular, we impose the key physical restraint that drifting snow is required for near-surface compaction in addition to overburden stress compaction. This mechanism has previously been proposed in the literature (Brun et al., 1997) and recently confirmed in both laboratory experiments (Sommer et al., 2017) as well as field observations on the AIS (Sommer et al., 2018). In our scheme, new snowfall is assigned a typically low (30 - 150 $\text{kg m}^{-3}$) density $\rho_{\text{new}}$, Eq. (1), according to the default SNOWPACK alpine new snow density parameterization which is a multiple linear regression derived from observations in the Swiss Alps (Lehning et al., 2002a), and varies as a function of 2 m air temperature $T_{2m}$ (°C), snow surface temperature $T_s$ (°C), relative humidity $RH$ (0 - 1), and 10 m wind speed $U_{10\text{ m}}$ ($\text{m s}^{-1}$).

$$\rho_{\text{new}} = 70 + 6.5T_{2m} + 7.5T_s + 0.26RH + 13U_{10\text{ m}} - 4.5T_{2m}T_s - 0.65T_{2m}U_{10\text{ m}} - 0.17RHU_{10\text{ m}} + 0.06T_{2m}T_sRH \qquad (1)$$

Densification then occurs via two distinct processes, a) densification due to overburden stress and b) densification due to drifting snow compaction. Densification due to overburden stress is calculated nearly identically as described in Steger et al. (2017), which adapted SNOWPACK for use in the percolation zone of the Greenland Ice Sheet by tuning the viscosity parameters including the snow activation energy $Q_s$ and critical exponent $\beta$. Here, we find that the parameter tuning proposed by Steger et al. (2017) leads to significantly overestimated densities (bias > 50 kg m$^{-3}$) in the dry snow zone of Antarctica. Therefore we revert to original SNOWPACK viscosity parameters by setting $Q_s$ and $\beta$ to 67,000 J mol$^{-1}$ and 0.7, respectively. The second mechanism driving densification, i.e. drifting snow compaction, occurs when the friction velocity $u_*$ (m s$^{-1}$) exceeds the snow threshold friction velocity $u_{*\mathrm{th}}$ (m s$^{-1}$), the minimum friction velocity at which surface snow grains are mobilized by the wind (Fig. 1). In our implementation of SNOWPACK, $u_*$ is estimated by scaling hourly averaged 10 m wind speeds from the MERRA-2 atmospheric reanalysis (section 2.2) using a logarithmic wind profile and stability corrections (Michlmayr et al., 2008) with a roughness length, $z_0$, of 2 mm. Meanwhile, $u_{*\mathrm{th}}$, Eq. (2), is calculated as a function of of snow microstructural properties internally determined by SNOWPACK, including snow grain sphericity $SP$ (0 - 1), radius $r_g$ (m), bond radius $r_b$ (m), and coordination number $N_3$ (Lehning and Fierz, 2008).

$$u_{*\mathrm{th}} = \sqrt{\frac{A\rho_i g r_g (SP+1) + B\sigma N_3 \frac{r_b^2}{r_g^2}}{\rho_a}} \tag{2}$$

In Eq. (2), $\rho_i$ is the density of ice (917 kg m$^{-3}$), $\rho_a$ is the density of air (1.1 kg m$^{-3}$), $g$ is the gravitational acceleration (9.8 m s$^{-2}$), $\sigma$ is a reference shear strength set to 300 $Pa$, while constants $A$ and $B$ are set to 0.02 and 0.0015 respectively. When $u_*$ exceeds $u_{*\mathrm{th}}$, a saltation mass transport rate $Q$ (kg m$^{-1}$ s$^{-1}$) is calculated following Lehning and Fierz (2008) and then scaled to a saltation mass flux $\Phi$ (kg m$^{-2}$ s$^{-1}$), Eq. (3), by dividing $Q$ by a characteristic horizontal length scale $L$.

$$\Phi = \frac{Q}{L} = \frac{0.0014\rho_a u_* (u_* - u_{*\mathrm{th}})(u_* + 7.6u_{*\mathrm{th}} + 205)}{L} \tag{3}$$

$L$ can be interpreted as a fetch length and characteristic horizontal length scale over which the originally upwind and now mobilized snow particles, which make up the saltation mass flux $\Phi$, have been eroded from the snow surface. We choose a magnitude for $L$ of 10 m, as this is a typical horizontal length scale of wind erosion features including sastrugi and barchan dunes (Filhol and Sturm, 2015). Given a lack of direct observations, $L$ can effectively be considered a tuning parameter, whose magnitude is inversely proportional to $\Phi$. Note that as suspension of drifting snow is not considered in the saltation model (Lehning and Fierz, 2008), the saltation mass transport rate $Q$ may underestimate the total mass flux in the saltation and suspension layers. However, the scaling of the mass transport rate by the poorly constrained fetch length $L$ to obtain the saltation mass flux $\Phi$ could be considered a way to include both suspension and saltation in the simulations.

Following erosion, the drifting snow mass flux is redeposited at the snow surface with an updated density $\rho_{\mathrm{redeposit}}$ (kg m$^{-3}$), Eq. (4), parameterized according to field measurements of surface snow deposited during drifting snow events (Groot Zwaaftink et al., 2013, equation 1). Note that, in contrast to Groot Zwaaftink et al. (2013), which uses 100-hour average wind speeds, we

calculate $\rho_{\text{redeposit}}$ using hourly mean MERRA-2 10 m wind speeds. We implement this distinction in order to better resolve the temporal evolution of redeposited snow density during ephemeral (i.e. shorter than 100 hours) drifting snow events.

$$\rho_{\text{redeposit}} = \begin{cases} 361 \log_{10}(U_{10\ \text{m}}) + 33 & U_{10\ \text{m}} > 1\ \text{m s}^{-1} \\ 33 & U_{10\ \text{m}} \leq 1\ \text{m s}^{-1} \end{cases} \tag{4}$$

## 2.2 SNOWPACK atmospheric forcing

At the snow surface, we force SNOWPACK with 1980 – 2017 MERRA-2 global atmospheric reanalysis (Gelaro et al., 2017) hourly mean 2 m air temperature, relative humidity, 10 m wind speed, incoming shortwave and longwave radiation (ISWR and ILWR), and precipitation rate. At the bottom of the simulated snowpack, we apply the MERRA-2 1980 – 2017 mean annual surface temperature as a thermodynamic boundary condition. We choose MERRA-2 because it provides a low release latency (approximately one month) and publicly available description of the state of the atmosphere, whereas regional climate models are not always regularly updated. This advantage, although not strictly necessary for this study as we focus on a past period (1980 - 2017), would be advantageous for future applications where timely estimates of snow properties are required, for example rapid interpretation of satellite imagery and altimetry, or determining the status of field assets (e.g. weather stations). Additionally, Medley and Thomas (2019) reported MERRA-2 to have the lowest accumulation bias compared to observations among comparable reanalysis products. Furthermore, Gossart et al. (2019) reports generally good agreement between MERRA-2 and observed near-surface climatology, including 2 m air temperature and wind speeds, however MERRA-2 appears to overestimate SMB, with an ice sheet wide mean absolute error of 58.5 $\text{kg m}^{-2}\,\text{yr}^{-1}$. According to Gossart et al. (2019), MERRA-2 well captures coastal and ice shelf SMB but generally overestimates SMB in the escarpment zone and at elevations from 500 - 3000 m.

We evaluate MERRA-2 atmospheric reanalysis as forcing for SNOWPACK by comparing with monthly averaged observations at nine automatic weather stations (AWSs), whose locations are shown in Figure 2. Note that in contrast to Gossart et al. (2019), our meteorological forcing evaluation relies on AWSs located primarily in Dronning Maud Land, and therefore may not be representative of the diverse range of Antarctic surface climates. By evaluating meteorological forcing at monthly timescales, we determine if there are any consistent biases that can impact simulated firn density and temperature profiles. However, it also smooths out high frequency discrepancies that may be important when evaluating instantaneous simulated density profiles. The AWSs measure temperature, relative humidity, wind speed and direction, air pressure, and the full radiation balance, i.e. incoming and reflected short wave radiation, and incoming and outgoing longwave radiation. Data from these stations are presented in Reijmer (2002) and Jakobs et al. (2020) and have been previously used to evaluate remote sensing retrievals (Trusel et al., 2013), ice core paleoclimate records (Medley et al., 2018), and climate models (van Wessem et al., 2018). According to our analysis and consistent with the findings of Lenaerts et al. (2017) and Gossart et al. (2019), MERRA-2 well captures observed 2 m air temperature, relative humidity, and wind speed, but significantly underestimates both ISWR and ILWR. We calculated an average MERRA-2 bias across all nine AWSs of -15.1 $\text{W m}^{-2}$ (corresponding to 19.4%) and -16.9 $\text{W m}^{-2}$ for ISWR and ILWR, respectively (Fig. A1). In order to reduce this bias in incoming radiation and thus better

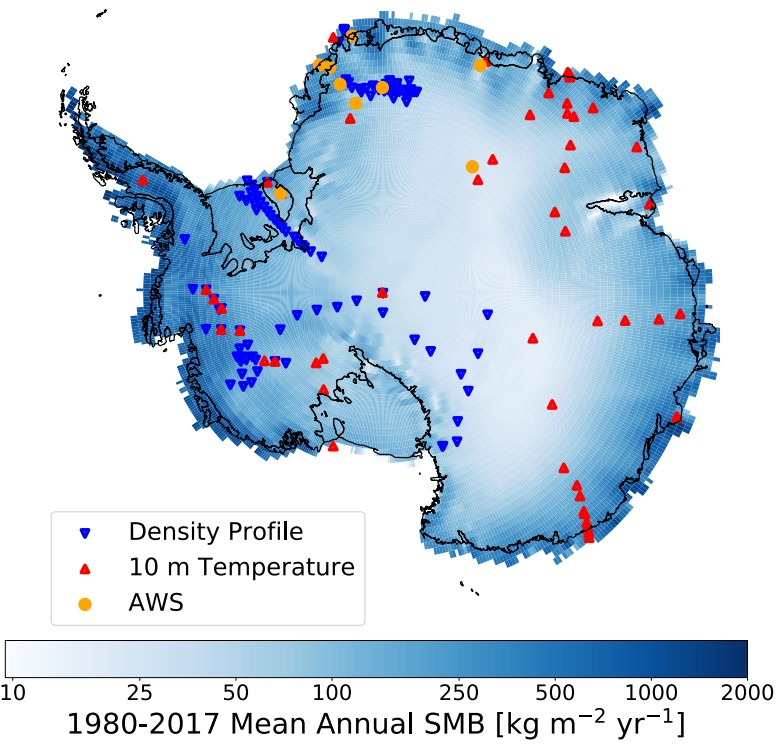

**Figure 2. Map of SNOWPACK simulation locations over the Antarctic ice sheet.** SNOWPACK simulation locations at 122 observed density profiles (upside down blue triangles), 55 borehole 10 m depth temperature measurements (red triangles), and nine automatic weather stations (AWS, yellow circles) plotted over MERRA-2 1980 - 2017 mean annual SMB.

capture AWSs observations, we increase MERRA-2 ISWR by 19.4 % and ILWR by 16.9 $\mathrm{W\,m^{-2}}$. Note that we choose a multiplicative correction for ISWR because a constant increase would be inappropriate when ISWR is low or zero, for example during twilight or polar night. Following bias correction, MERRA-2 ISWR and ILWR biases are reduced to 8.7 $\mathrm{W\,m^{-2}}$ and -0.6 $\mathrm{W\,m^{-2}}$, respectively. Additionally, root mean squared error (RMSE) ISWR is reduced from 22.7 $\mathrm{W\,m^{-2}}$ to 20.3 $\mathrm{W\,m^{-2}}$ while ILWR RMSE is reduced from 21.5 $\mathrm{W\,m^{-2}}$ to 14.1 $\mathrm{W\,m^{-2}}$. Note that this bias correction only impacts ISWR and ILWR forcing, not 2 m air temperature, relative humidity, 10 m wind speed, and precipitation rate. Additionally, for the rest of this study, all simulations are driven by bias corrected ISWR and ILWR, unless explicitly stated otherwise.

### 2.3   SNOWPACK model initialization

In our scheme, new snow layers are added on top of the snow column when precipitation is prescribed by the atmospheric forcing, in steps of 2 cm. Layers are initialized with a density given by Eq. 1 when they originate from precipitation. When snow layers are eroded and redeposited, the layers originating from drifting snow are initialized with a density given by Eq. 4. Initial grain size for all newly added layers is 0.2 mm (Groot Zwaaftink et al., 2013). There are two sets of microstructural

properties for grain shape (dendricity and sphericity), for high and low wind speed, respectively (Groot Zwaaftink et al., 2013). Note that precipitation is treated before assessing snow erosion, such that low density snow from precipitation can erode immediately when conditions allow. To reduce computational costs, a sophisticated snow layer merging scheme merges layers with very low ice content due to sublimation or melt and layers with similar properties (density, water content, grain size, and grain shape parameters). The criteria for layer merging are relaxed with depth, to allow for more aggressive layer merging. At 10 m depth, typical layer spacing is around 10 - 20 cm. Near the surface, layers can be split to maintain a vertical resolution of a few cm in order to numerically represent steep temperature gradients.

In order to ensure a realistic representation of snow and firn properties throughout the entire near-surface, we complete a SNOWPACK model spinup such that simulated snow depth is 10 m at all sites before comparison with observations or other SNOWPACK simulations. We choose 10 m in order to resolve seasonal variability in snow and firn characteristics as well as energy exchange with the atmosphere. For spinup, we mimic the method presented in Ligtenberg et al. (2011) by repeating the 1980 – 2017 model period until initial simulated snow depth on January 1st, 1980 is at least 10 m. Once spinup is complete, we perform one final 1980 – 2017 model simulation, which we use for the analysis.

### 2.4 Borehole 10 m temperature as mean annual surface temperature proxy

To test MERRA-2 and SNOWPACK's ability to capture the surface energy balance (SEB) across a range of AIS surface climates, we compare 1980 – 2017 mean MERRA-2 surface temperature and SNOWPACK snow surface temperature with 10 m depth temperatures from 55 boreholes whose locations are show in Fig. 2. In the absence of significant surface meltwater percolation, 10 m depth temperature equilibrates with mean annual surface temperature, and can therefore be used as a proxy for surface temperature and by extension SEB in the absence of direct observations (van den Broeke, 2008; Lenaerts et al., 2012; van Wessem et al., 2014).

### 2.5 IMAU-FDM model

To aid in the evaluation of SNOWPACK modeled near-surface density, we also compare with the widely used Institute for Marine and Atmospheric research Utrecht firn densification model (IMAU-FDM v1.1) (Ligtenberg et al., 2011; Kuipers Munneke et al., 2015; Ligtenberg et al., 2018). The IMAU-FDM is a semi-empirical firn densification model designed to represent snow and firn cover processes including densification, meltwater refreezing and percolation, and surface height change. Gridded IMAU-FDM density profiles are available at 27 km horizontal, 4 cm vertical, and 30 day temporal resolution with atmospheric forcing provided by the regional climate model RACMO 2.3p2 (van Wessem et al., 2018). In contrast to SNOWPACK, which relies on an overburden stress compaction scheme, IMAU-FDM uses a calibrated semi-empirical dry snow densification scheme based on Arthern et al. (2010), which describes densification as a function of density as well as annual average accumulation and temperature. In further contrast to SNOWPACK, the IMAU-FDM parameterizes new snow density as a function of annual average, rather than hourly, meteorology and currently includes no post-deposition mechanism to increase surface snow density due to drifting snow processes.

## 2.6 GSFC-FDM model

In addition to IMAU-FDM we also compare SNOWPACK to the NASA Goddard Space Flight Center firn densification model
(GSFC-FDMv1) which provides simulated firn properties over the past 40 years (1980 - 2019) for both the Greenland and
Antarctic ice sheets (Medley et al., 2020). GSFC-FDM uses the Community Firn Model, a modular, open-source framework
for Lagrangian modeling of several firn and firn-air related processes within a single column (Stevens et al., 2020). The GSFC-
FDM simulations are forced by an enhanced resolution hybridized MERRA-2 that was developed by exploiting a 15-year,
12.5 km resolution offline MERRA-2 reanalysis. The hybridized MERRA-2 forcing retains the spatial gradients in the high-
resolution reanalysis while maintaining the temporal variations from the original MERRA-2 variables (see Medley et al. (2020)
for complete details). The dry snow and firn compaction model, based on Arthern et al. (2010), was calibrated to observed
depth-density profiles from both Greenland and Antarctica. A simple initial density scheme was implemented based on mean
annual MERRA-2 climate, which provides a spatially variable initial density that does not, in contrast to SNOWPACK, vary
in time or vary due to drifting snow processes.

## 2.7 SNOWPACK, IMAU-FDM, and GSFC-FDM comparison with density observations

We evaluate skill for all three firn models in representing surface and near-surface snow and firn density, defined here as the
average density between depths of $0 - 1$ m and $0 - 10$ m, respectively, by comparing with community sourced and publicly
available density profiles from the Surface Mass Balance and Snow on Sea Ice Working Group (SUMup) data set (Albert, 2007;
Medley et al., 2013; Montgomery et al., 2018). We first identified all available Antarctic SUMup density observations in the
near-surface and then retrieved spatially and temporally consistent SNOWPACK, IMAU-FDM, and GSFC-FDM model output.
For all models, we retrieve the simulated density profile whose time step is closest to that of the observed profile. SNOWPACK,
IMAU-FDM, and GSFC-FDM report simulated density profiles every 1, 30, and 5 days respectively. This classification yielded
122 unique observed profiles (Fig. 2) that are located primarily on the grounded ice sheet, where surface melt is limited ($<$ 50
mm w.e. $\mathrm{yr}^{-1}$), absent, or very rare (Trusel et al., 2013). Observations and models report different vertical resolution density,
therefore we compute the mean and standard deviation of all 122 density profiles at ten $1$ m thick vertical levels, beginning at
$0 - 1$ m and ending with $9 - 10$ m depths.

## 3 Results and Discussion

### 3.1 Surface energy balance

The large variability in observed $10$ m depth borehole temperatures, ranging from -57.0 to -14.4 °C, is well captured by
MERRA-2, non-bias corrected SNOWPACK, and bias corrected SNOWPACK, indicating proper representation of the average
SEB (Fig. 3). In particular, the R-squared values and slope of linear regressions are 0.96 and 1.00 for MERRA-2, 0.97 and 0.98
for non-bias corrected SNOWPACK, and 0.97 and 0.95 for bias corrected SNOWPACK, respectively. MERRA-2 overestimates
$T_s$, with an average bias of 0.52 °C, while non-bias corrected SNOWPACK underestimates $T_s$ with an average bias of -0.99

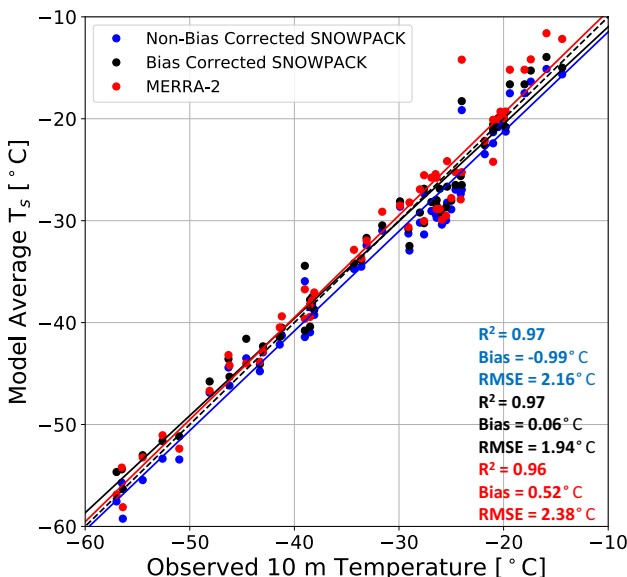

**Figure 3. Modeled surface temperature evaluation.** Comparison between observed borehole 10 m depth temperature (°C) and 1980 - 2017 model average non-bias corrected SNOWPACK (blue), bias corrected SNOWPACK (black), and MERRA-2 (red) surface temperature $T_s$ (°C). The dashed black line represents a one to one line. Solid lines represent linear regressions. R-squared values, mean bias, and RMSE are reported for non-bias corrected SNOWPACK (blue text), bias corrected SNOWPACK (black text), and MERRA-2 (red text).

°C. Meanwhile, bias corrected SNOWPACK yields excellent agreement, with an average bias of 0.06 °C. MERRA-2 shows
the largest RMSE, 2.38 °C, while the non-bias corrected and bias corrected versions of SNOWPACK display an RMSE of 2.16 and 1.94 °C, respectively.

Since the MERRA-2 bias in ISWR and ILWR is determined using only 9 AWSs located primarily in Dronning Maud Land (section 2.2), we were initially concerned with the bias' spatial representativeness. However, following bias correction, the statistically significant reduction in both RMSE and bias (p < 0.01) magnitude when compared to 10 m depth temperatures,
indicates a clear improvement in SNOWPACK's representation of Antarctic SEB, including locations outside of Dronning Maud Land.

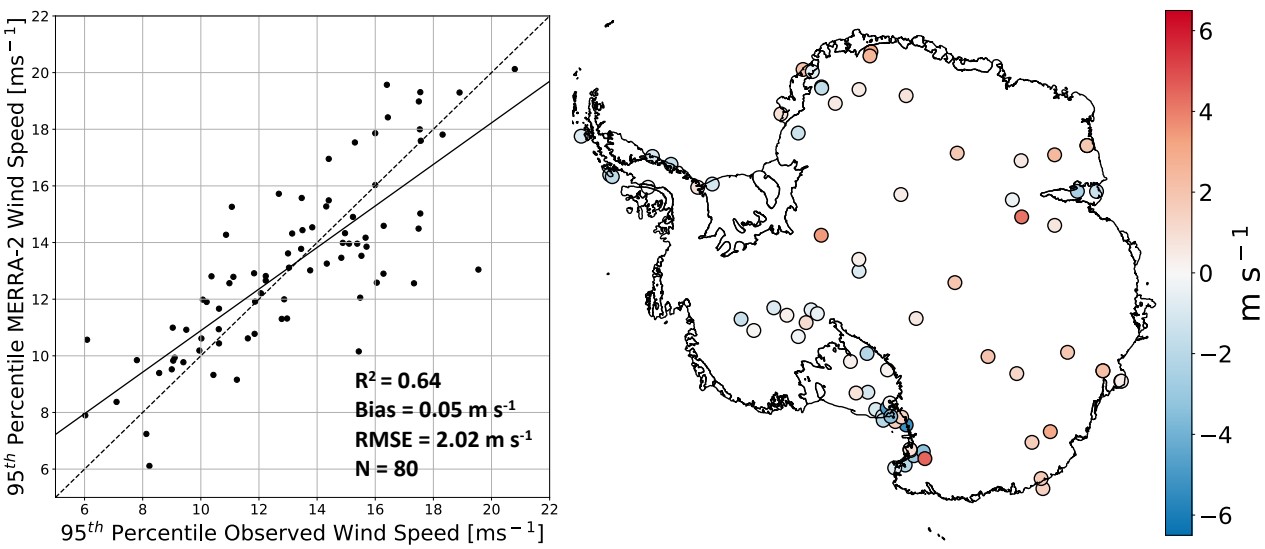

**Figure 4. Evaluation of MERRA-2 $95^{th}$ percentile hourly wind speeds.** Scatter plot (left) of observed vs. MERRA-2 simulated $95^{th}$ percentile hourly 10 m wind speeds at 80 automatic weather stations scattered across the Antarctic ice sheet. Map of MERRA-2 minus observed $95^{th}$ percentile hourly wind speeds (right).

## 3.2 Evaluation of MERRA-2 $95^{th}$ percentile hourly wind speeds

Accurate model representation of drifting snow frequency and intensity, as well as the density of drifting snow accumulation, relies on a realistic representation of the strongest wind events. Although Gossart et al. (2019) report good agreement between annual mean observed and MERRA-2 simulated wind speeds, this finding does not necessarily apply to the strongest wind events, when drifting snow is most significant. Therefore, to better understand the quality of MERRA-2 atmospheric forcing during strong drifting snow events, we compare observed and MERRA-2 simulated $95^{th}$ percentile hourly 10 m wind speeds at 80 AWSs across the AIS (Sanz Rodrigo et al., 2013). MERRA-2 demonstrates relatively good model performance ($R^2$ = 0.64, bias = 0.05 $\mathrm{m\,s^{-1}}$, $RMSE = 2.02\ \mathrm{m\,s^{-1}}$, Fig. 4) and despite occasional (5 out of 80 AWSs) considerable differences (magnitude $> 4\ \mathrm{m\,s^{-1}}$), we see no systematic errors. Furthermore, MERRA-2 shows no appreciable spatial pattern in bias, for example in the ice sheet interior or coast. Thus, we conclude that MERRA-2 demonstrates considerable skill in representing the 5% strongest hourly wind speeds and can therefore be used to reliably estimate the friction velocity $u_*$ and saltation mass flux $\Phi$, Eq. (3).

## 3.3 SNOWPACK density profiles sensitivity to atmospheric forcing uncertainty.

Although we have performed an evaluation of MERRA-2 SEB, and introduced a bias correction for ISWR and ILWR, uncertainties in bias-corrected MERRA-2 atmospheric forcing likely still exist, and can impact simulated density profiles. To

understand the effect of these uncertainties on SNOWPACK simulated density, we perform an ensemble of SNOWPACK simulations at South Pole and the West Antarctic Ice Sheet divide (WAIS), where we perturb MERRA-2 prescribed 10 m wind speed, 2 m air temperature, and precipitation. We choose South Pole (90°S 0°E) and WAIS divide (79.5°S 112°W) because these are well known points of interest and represent distinct climatic and accumulation regimes with mean annual surface temperatures of -52.4 and -29.4 °C, and annual snow accumulation of 56 and 207 $\mathrm{kg\,m^{-2}\,yr^{-1}}$ respectively. Because MERRA-2 exhibits mean absolute errors of 2.4 $\mathrm{m\,s^{-1}}$ and 3.1 °C for annual average 10 m wind speeds and near-surface temperature (Gossart et al., 2019) and typical relative accumulation errors of 20 % (Medley and Thomas, 2019), we independently increase and decrease MERRA-2 10 m wind speed, 2 m air temperature, and precipitation at South Pole and WAIS by 2.4 $\mathrm{m\,s^{-1}}$, 3.1 °C, and 20 %, respectively.

SNOWPACK simulated density sensitivities to uncertainties in 10 m wind speed are considerably larger than that of 2 m air temperature and precipitation and range from -46.6 $\mathrm{kg\,m^{-3}}$ at 0 - 1 m in the South Pole - 2.4 $\mathrm{m\,s^{-1}}$ simulation to 58.5 $\mathrm{kg\,m^{-3}}$ at 2 - 3 m in the WAIS + 2.4 $\mathrm{m\,s^{-1}}$ simulation (Fig. 5). Meanwhile, across our 2 m air temperature and precipitation ensemble, differences in simulated density are generally less than 5 % of non-perturbed density in 75 out of 80 cases, and never exceed 10 %. Because these simulated density differences are typically small compared to absolute density, we conclude that uncertainties arising due to 2 m air temperature and precipitation alone are smaller in magnitude than uncertainties arising from firn densification model choice (sections 3.5 - 3.7). On the other hand, density differences in the 10 m wind speed experiments are larger, particularly from 0 - 3 m depth, and exceed 5 % of non-perturbed density in 24 out of 40 cases. Thus, a realistic representation of wind speed is crucial for simulating realistic density profiles. However, in the context of our study, simulated density uncertainties arising from uncertainties in MERRA-2 wind speed cannot be easily reduced, because, consistent with Gossart et al. (2019), we find no substantial bias in MERRA-2 wind speed compared to observations (section 3.2). We must therefore acknowledge that density variations due to uncertainty in wind speed represent the largest source of uncertainty with regard to SNOWPACK simulated near-surface density, and in fact, exceeds uncertainties arising from firn densification model choice.

## 3.4 Effect of new SNOWPACK drifting snow compaction routine on simulated density profiles

In earlier studies, SNOWPACK described enhanced surface compaction under the influence of wind by increasing the strain rate of snow in the top 7 cm (Groot Zwaaftink et al., 2013). For simulations in Antarctica, a higher strain rate was proposed when compared to simulations of seasonal snow cover in alpine terrain. Alternatively, the new drifting snow compaction scheme we have introduced in this study does not modify the strain rate. Instead, surface compaction by wind is only introduced when the friction velocity is sufficient to first erode, and then subsequently redeposit snow with a higher density. To demonstrate the influence of our new drifting snow compaction scheme on simulated near-surface density profiles, we have plotted December 30th, 1997 observed and SNOWPACK simulated density profiles at Kohnen Station in Dronning Maud Land, East Antarctica (75°S, 0°E, 2891 m, Fig. 6). For SNOWPACK, we have included three sets of model physics 1) this study, denoted "Redeposit", 2) the wind-enhanced surface compaction scheme using the strain rate proposed by Groot Zwaaftink et al. (2013), denoted

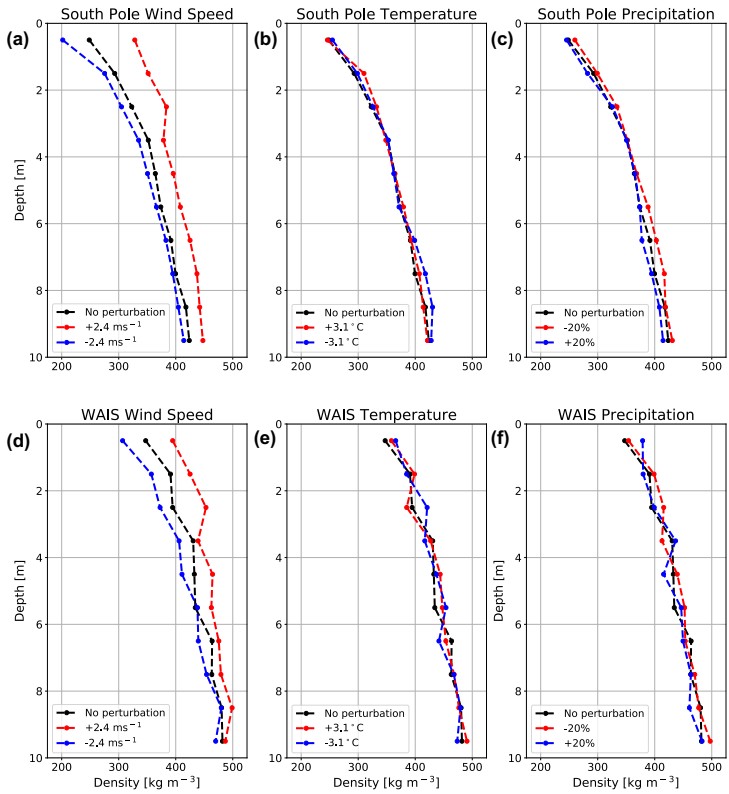

**Figure 5. SNOWPACK simulated density profile sensitivity to uncertainties in atmospheric forcing.** SNOWPACK simulated density profile sensitivity at South Pole (a,b,c) and WAIS (d,e,f), to wind speed (a,d), temperature (b,e), and precipitation (c,f). The increased wind speed (+ 2.4 $\mathrm{m\,s^{-1}}$), increased temperature (+ 3.1 °C), and reduced precipitation (- 20 %) perturbations are shown in red, while decreased wind speed (- 2.4 $\mathrm{m\,s^{-1}}$), decreased temperature (- 3.1 °C), and increased precipitation (+ 20 %) perturbations are shown in blue. In all panels, density profiles are valid for December 31st, 2017 and black curves represent unperturbed atmospheric forcing simulations.

"Enhanced Compaction", and 3) the default alpine SNOWPACK snow physics, which includes only a modest wind-enhanced surface compaction using the strain rate (Bartelt and Lehning, 2002; Lehning et al., 2002a, b), denoted "Default".

     From our analysis, it is immediately clear that the default SNOWPACK snow physics significantly underestimate observed surface density ($> 150 \mathrm{\ kg\,m^{-3}}$), suggesting that the wind effect described in the new snow density parameterization (Eq. 1) and the increased strain rate in this variant are insufficient for drifting-snow dominated polar environments. However, in

the case of "Default" snow physics, this underestimation of density decreases with depth, which can be explained by the reduced viscosity of relatively lower density snow, and therefore more rapid densification due to overburden stress. For the "Enhanced Compaction" simulation, we see similar behavior, however modeled surface density errors are reduced compared to the "Default" simulation. Compared to "Redeposit", the "Enhanced Compaction" simulation significantly underestimates

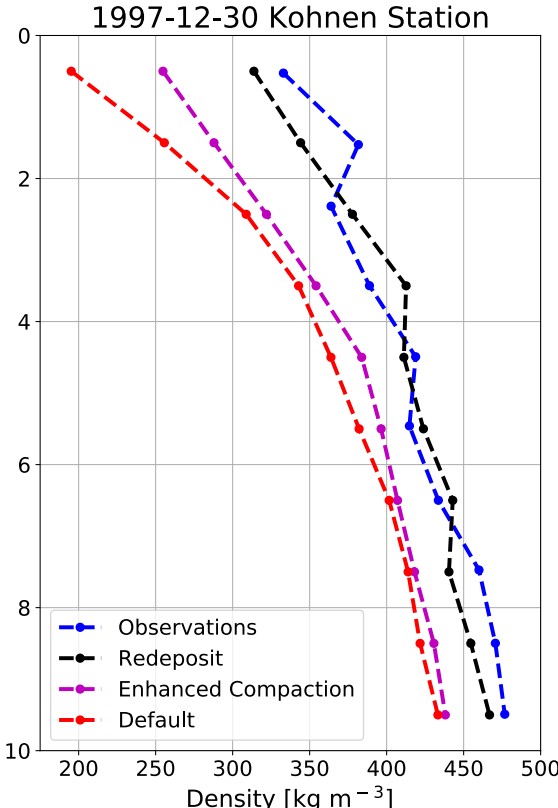

**Figure 6. SNOWPACK simulated density profiles sensitivity to prescribed snow physics.** Observed (blue), SNOWPACK simulated "Redeposit" (black), "Enhanced Compaction" (magenta), and "Default" (red) near-surface density profiles at Kohnen Station in Dronning Maud Land (75.0°S, 0.0°W, 2891 m).

near-surface density. This demonstrates that the enhanced wind compaction mechanism via the strain rate is not sufficient to capture observations. Instead, describing compaction of surface snow due to wind using wind-driven erosion and subsequent redeposition at a higher density ("Redeposit"), which resembles the experimentally determined mechanism (Sommer et al., 2017), adequately captures observed near-surface density.

### 3.5 Surface snow density

A comparison between both SNOWPACK, IMAU-FDM, and GSFC-FDM simulated surface snow density, defined as the top meter, with 79 unique observations is shown in Figure 7. Observed densities range from $272 - 507 \, \mathrm{kg\,m^{-3}}$, with a mean of 362 $\mathrm{kg\,m^{-3}}$ and standard deviation of $39 \, \mathrm{kg\,m^{-3}}$. Since it is known that meteorological conditions including annual accumulation and temperature influence Antarctic snow and firn density (Herron and Langway, 1980), we tested this hypothesis. We find a modest relationship between MERRA-2 1980 – 2017 mean annual SMB and observed surface snow density ($p < 0.01$, $R^2$

= 0.23) as well as a significant, but weak correlation between elevation and observed surface snow density (p < 0.01, $R^2$ = 0.16), likely due to the relationship between elevation and surface climate. Note that we find no significant correlation between MERRA-2 1980 – 2017 mean wind speed and observed surface density (p = 0.14, $R^2$ = 0.03) but do in fact find a significant, albeit weak relationship between 1980 – 2017 maximum wind speed and surface density (p < 0.01, $R^2$ = 0.09). Because drifting snow compaction is known to partially control snow density on daily to hourly timescales (Sommer et al., 2018), the lack of a significant relationship between mean annual wind speed and surface snow density combined with the significant relationship between maximum wind speed and surface density indicates the importance of resolving drifting snow compaction with high temporal resolution (daily to hourly) meteorological forcing as opposed to annual means or climatology.

SNOWPACK (linear fit slope = 0.51, $R^2$ = 0.20), IMAU-FDM (linear fit slope = 0.20, $R^2$ = 0.19), and GSFC-FDM (linear fit slope = 0.22, $R^2$ = 0.36) all modestly capture observed surface snow density variability. Both SNOWPACK and IMAU-FDM underestimate surface snow density with an average bias of -8.2 and -20.4 $\mathrm{kg\,m^{-3}}$ and RMSE of 45.3 and 40.7 $\mathrm{kg\,m^{-3}}$, respectively. Meanwhile GSFC-FDM, on average overestimates surface snow density, with a bias of 20.4 $\mathrm{kg\,m^{-3}}$ and RMSE of 38.5 $\mathrm{kg\,m^{-3}}$. For all models, we identify no clear relationship between geographic location and model bias. However, we find a significant but weak negative relationship between observed density and modeled surface density bias for SNOWPACK (p < 0.01, $R^2$ = 0.19) as well as a moderately strong negative relationship for IMAU-FDM (p < 0.01, $R^2$ = 0.80) and strong negative relationship for GSFC-FDM (p < 0.01, $R^2$ = 0.87).

According to our analysis, SNOWPACK better captures the range in observed surface density when compared to IMAU-FDM, as evidenced by a lower magnitude bias, higher $R^2$, and a linear fit slope closer to unity, but the typical error is larger, as expressed by a slightly larger RMSE (Table 1). Likewise, when compared to GSFC-FDM, SNOWPACK exhibits a lower surface density bias magnitude and has a linear slope closer to unity, while it shows a larger RMSE and lower $R^2$. Additionally, we calculate a p-value of 0.06 for a two-sided t-test in which the null hypothesis states that SNOWPACK and IMAU-FDM have identical average biases. This information combined with SNOWPACK's larger RMSE compared to IMAU-FDM and GSFC-FDM, and smaller $R^2$ compared to GSFC-FDM, leads us to conclude that neither model is superior, and instead that all three models perform comparably with regard to surface density. Note that all three models perform particularly poorly when observed surface density exceeds 400 $\mathrm{kg\,m^{-3}}$ (SNOWPACK: Bias = -23.7 $\mathrm{kg\,m^{-3}}$, RMSE = 65.8 $\mathrm{kg\,m^{-3}}$; IMAU-FDM: Bias = -65.4 $\mathrm{kg\,m^{-3}}$, RMSE = 74.7 $\mathrm{kg\,m^{-3}}$; GSFC-FDM: Bias = -20.4 $\mathrm{kg\,m^{-3}}$, RMSE = 35.6 $\mathrm{kg\,m^{-3}}$). The origin of this poor performance at high observed surface densities is still unknown, but we can come up with at least three potential explanations. First, we speculate that high surface densities are found in areas with locally steep topography. However, we find no significant relationship between surface slope derived from a 1 $\mathrm{km}$ horizontal resolution digital elevation model (Helm et al., 2014) and modeled surface density bias (not shown), so this mechanism is an unlikely candidate to explain the bias. Second, this model degradation at high observed surface densities could potentially be explained by an underestimation of new snow density and/or the frequency/intensity of simulated drifting snow compaction due to the presence of local meteorological phenomena not captured by MERRA-2 or RACMO2. Lastly, we cannot rule out the possibility of larger errors in the observational data for densities above 400 $\mathrm{kg\,m^{-3}}$. For example, some surface density observations, particularly those from deep firn and ice cores, may not be representative of an undisturbed snow surface and could therefore report an artificially high surface density.

Moreover, surface snow samples taken from cores are often compressed by the drill and can sometimes break into multiple
pieces, therefore confounding density measurements.

**Table 1.** SNOWPACK, GSFC-FDM, and IMAU-FDM simulated mean surface density bias and RMSE at all 79 sites and at sites whose observed surface density exceeds $400 \, \mathrm{kg \, m^{-3}}$.

|  | SNOWPACK | GSFC-FDM | IMAU-FDM |
|---|---|---|---|
| Bias ($\mathrm{kg \, m^{-3}}$) at all sites | -8.2 | 20.4 | -20.4 |
| Bias ($\mathrm{kg \, m^{-3}}$) at high surface density ($> 400 \, \mathrm{kg \, m^{-3}}$) sites | -23.7 | -20.4 | -65.4 |
| RMSE ($\mathrm{kg \, m^{-3}}$) at all sites | 45.3 | 38.5 | 40.7 |
| RMSE ($\mathrm{kg \, m^{-3}}$) at high surface density ($> 400 \, \mathrm{kg \, m^{-3}}$) sites | 65.8 | 35.6 | 74.7 |

## 3.6 Near-surface snow and firn densification

In a comparison with 122 observed density profiles, SNOWPACK exhibits a lower bias compared to IMAU-FDM for the entire near-surface, and a lower bias compared to GSFC FDM from $0 – 7 \, \mathrm{m}$ depth (Fig. 8). Observed mean densities range from 362 $\mathrm{kg \, m^{-3}}$ at depth $0 – 1 \, \mathrm{m}$ to $510 \, \mathrm{kg \, m^{-3}}$ at $9 – 10 \, \mathrm{m}$. Meanwhile, SNOWPACK mean densities range from $354 \, \mathrm{kg \, m^{-3}}$ at
355 depth $0 – 1 \, \mathrm{m}$ to $489 \, \mathrm{kg \, m^{-3}}$ at $9 – 10 \, \mathrm{m}$. SNOWPACK underestimates observed density with biases varying from a minimum of $-0.6 \, \mathrm{kg \, m^{-3}}$ at depth $2 – 3 \, \mathrm{m}$ to a maximum of $-25.3 \, \mathrm{kg \, m^{-3}}$ at $8 – 9 \, \mathrm{m}$. Note that the SNOWPACK mean density bias from $0 – 7 \, \mathrm{m}$ is typically small ($< 10 \, \mathrm{kg \, m^{-3}}$), and has approximately the same magnitude as density uncertainties that arise from uncertainties in atmospheric forcing (Section 3.3). The IMAU-FDM likewise underestimates near-surface density with biases ranging from $-20.4 \, \mathrm{kg \, m^{-3}}$ at depth $0 – 1 \, \mathrm{m}$ to $-40.1 \, \mathrm{kg \, m^{-3}}$ at $8 – 9 \, \mathrm{m}$. Meanwhile, GSFC-FDM overestimates near-
360 surface density with biases ranging from $8.5 \, \mathrm{kg \, m^{-3}}$ at depth $8 - 9 \, \mathrm{m}$ to $23.7 \, \mathrm{kg \, m^{-3}}$ at $2 – 3 \, \mathrm{m}$. Furthermore, SNOWPACK correctly predicts observed variability in mean density, as shown by similar standard deviations between SNOWPACK and observations (Fig. 8, panel A). Alternatively, the IMAU-FDM and GSFC-FDM both underestimate observed mean density variability at the surface, but converge towards the observed variability with increasing depth. Note that 95% confidence intervals on the SNOWPACK mean bias contain zero in the top $7 \, \mathrm{m}$ (Fig. 8), thus we conclude that SNOWPACK is consistent
with observations at these depths. However, the IMAU-FDM and GSFC-FDM mean bias 95% confidence intervals never contain zero and are therefore not consistent with observations. Despite these contrasting conclusions, SNOWPACK's mean bias confidence intervals occasionally overlap with that of the IMAU-FDM and GSFC-FDM (e.g. $0 – 1 \, \mathrm{m}$ and $8 – 10 \, \mathrm{m}$) in the case of IMAU-FDM) and therefore should not be considered uniformly statistically indistinguishable.

To test for the potential effect of compensating biases on our analysis of near-surface densification, we partition the 122 ob-
370 served density profiles into 35 high and 87 low SMB categories by applying a mean annual SMB threshold of $200 \, \mathrm{kg \, m^{-2} \, yr^{-1}}$ according to MERRA-2 (Fig. 9). We choose $200 \, \mathrm{kg \, m^{-2} \, yr^{-1}}$ because it roughly approximates the area weighted mean annual SMB over the grounded AIS ($172.8 \, \mathrm{kg \, m^{-2} \, yr^{-1}}$ according to Agosta et al. (2019)). Again, for both high and low SMB sites,

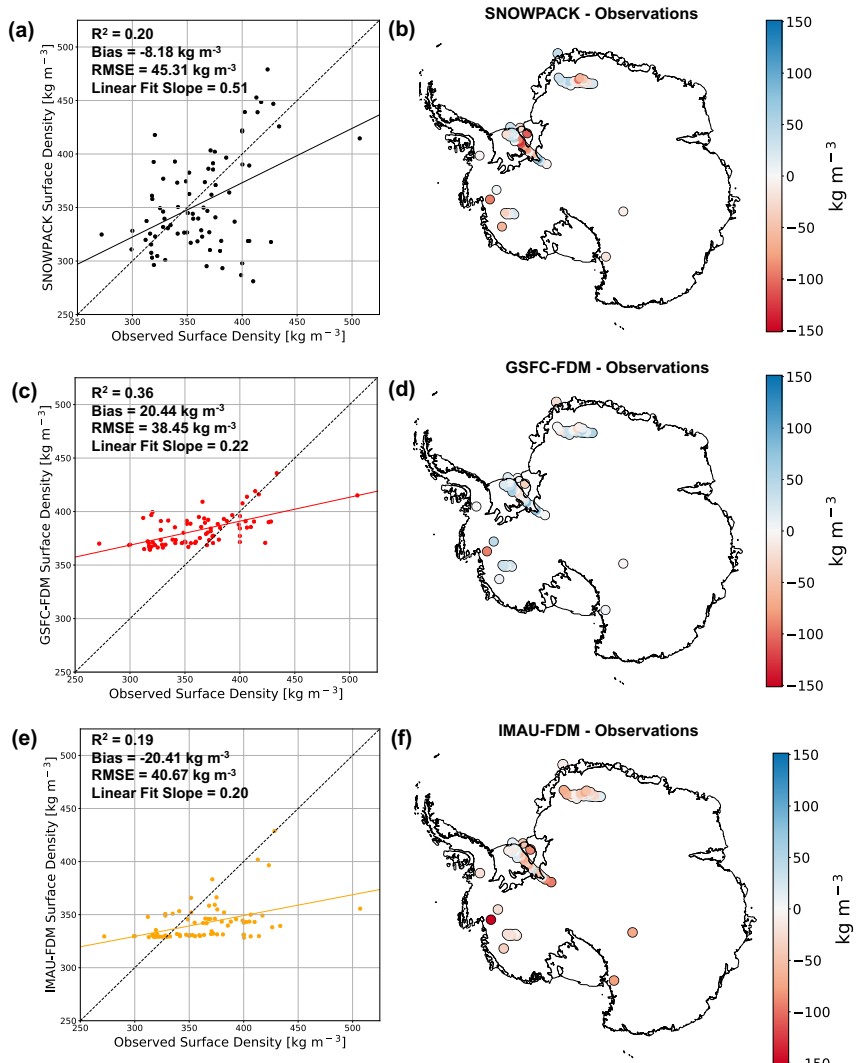

**Figure 7. Modeled and observed surface density comparison.** Scatter plots of observed vs. SNOWPACK (a), GSFC-FDM (c), and IMAU-FDM (e) modeled surface density at 79 locations across the AIS. Maps of SNOWPACK (b), GSFC-FDM (d), and IMAU-FDM (f) minus observed surface density at 79 locations across the AIS. In panels a, c, and e, dashed black lines represent one to one lines while solid lines represent linear regressions.

we find that SNOWPACK, when compared to IMAU-FDM, shows lower absolute density biases throughout the entire near-surface firn column, from the surface down to 10 m depth. Compared to GSFC-FDM, SNOWPACK exhibits a smaller absolute

near-surface density bias only from $0 - 8$ m and $9 - 10$ m at low accumulation sites. For high SMB sites, SNOWPACK mean biases range from -55.9 $\mathrm{kg\,m^{-3}}$ at $8 - 9$ m to -3.9 $\mathrm{kg\,m^{-3}}$ at $0 - 1$ m, while for low SMB sites, mean biases are generally

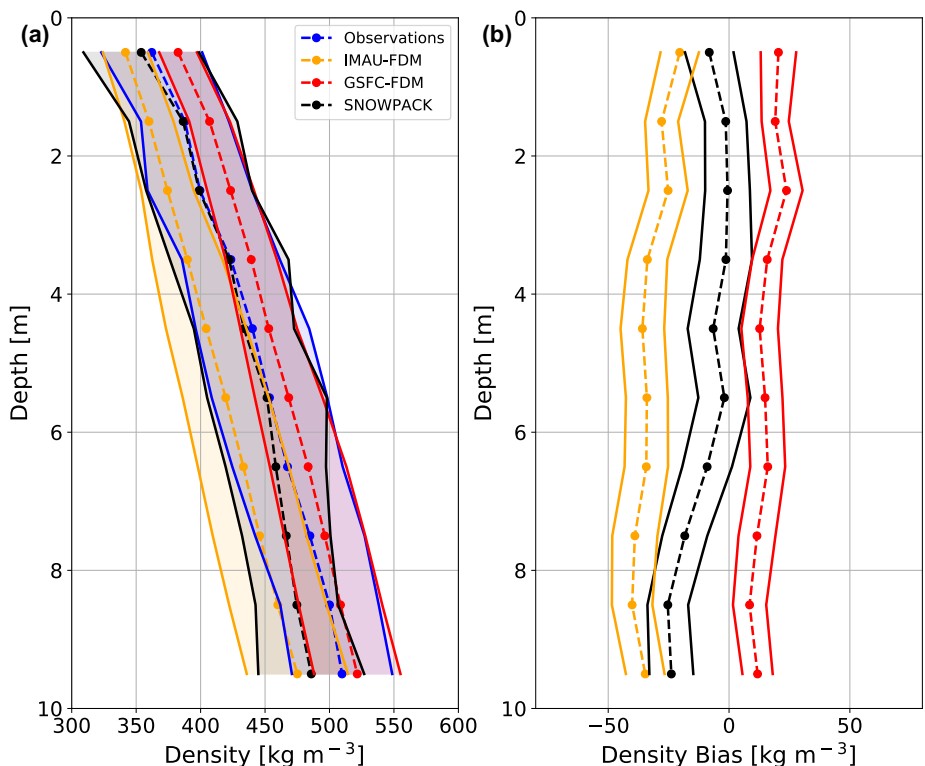

**Figure 8. Density profile comparison.** Mean observed (blue dashed), SNOWPACK modeled (black dashed), GSFC-FDM (red dashed) and IMAU-FDM modeled (yellow dashed) near-surface (0 - 10 m) density profiles at 122 locations across the AIS (a). Shading represents plus and minus one standard deviation across observed and modeled density profiles. Mean SNOWPACK minus observed (black dashed), GSFC-FDM minus observed (red dashed), and IMAU-FDM minus observed (yellow dashed) density profiles at 122 locations across the AIS (b). Error bars represent 95% confidence intervals on bias means.

reduced in magnitude and range from -14.3 $\mathrm{kg\,m^{-3}}$ at 9 – 10 m to 8.3 $\mathrm{kg\,m^{-3}}$ at 5 – 6 m. Given SNOWPACK's mean bias reduction, when compared to IMAU-FDM, and comparable peformance relative to GSFC-FDM, we have demonstrated our physics-based modeling approach is capable of reliably capturing Antarctic near-surface snow and firn density. Consistent with our analysis of all profiles in Figure 8, SNOWPACK, as measured by the mean bias 95% confidence intervals, is generally consistent with observations and statistically distinguishable from IMAU-FDM and GSFC-FDM at low accumulation sites. However, at high accumulation sites, SNOWPACK exhibits significantaly worse performance as indicated by a strongly negative bias, and confidence intervalls that do not contain zero, particularly below 6 m depth.

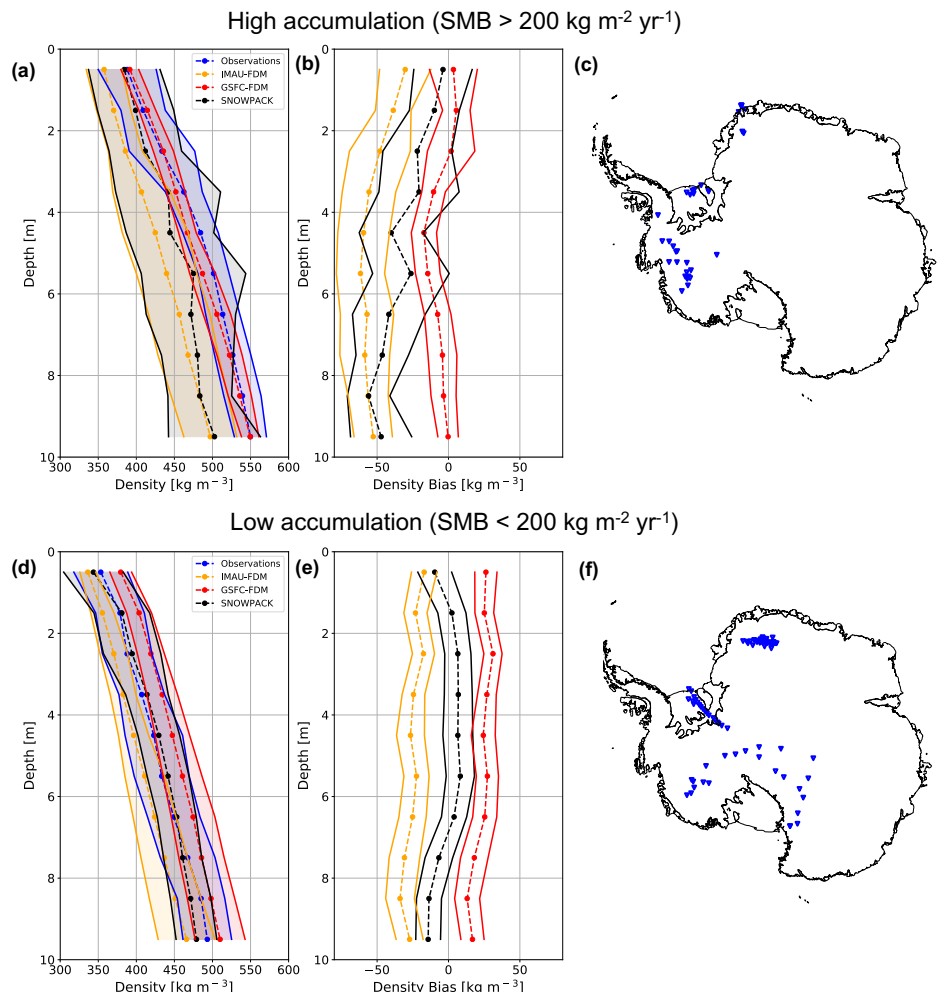

**Figure 9. High and low SMB density profile comparison.** Average observed (blue dashed), SNOWPACK modeled (black dashed), GSFC-FDM modeled (red dashed), and IMAU-FDM modeled (yellow dashed) near-surface (0 - 10 m) density profiles at high (a) and low (d) SMB sites. Shading represents plus and minus one standard deviation across observed and modeled density profiles. Mean SNOWPACK minus observed (black dashed), GSFC-FDM minus observed (red dashed), and IMAU-FDM minus observed (yellow dashed) density profiles at high (b) and low (e) SMB sites. Error bars represent 95% confidence intervals on bias means. Maps of locations of high (c) and low (f) SMB sites. High and low SMB sites are delineated by a $200 \, \mathrm{kg \, m^{-2} \, yr^{-1}}$ threshold applied to MERRA-2 1980 - 2017 mean annual SMB.

### 3.7 Simulated near-surface firn densification at sites not included in the calibration of IMAU-FDM and GSFC-FDM.

Semi-empirical firn densification models, including IMAU-FDM and GSFC-FDM, are calibrated against observed density profiles in order to improve model agreement with observations. However, because near-surface density observations are sparse, not evenly distributed in space, and typically collected in summer, semi-empirical models may be biased towards climate

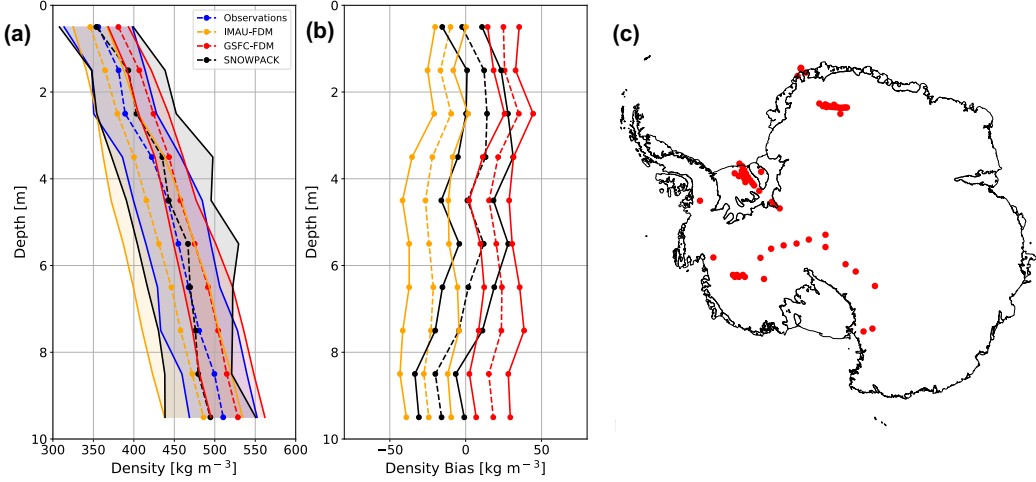

**Figure 10. SNOWPACK, IMAU-FDM, and GSFC-FDM density simulations at sites not included in the IMAU-FDM and GSFC-FDM calibrations.** Average observed (blue), SNOWPACK (black), IMAU-FDM (yellow), and GSFC-FDM modeled (red) near-surface (0 - 10 m) density profiles at 69 sites not included in the IMAU-FDM and GSFC-FDM density calibrations (a). Shading represents plus and minus one standard deviation across 69 observed and modeled density profiles. Mean SNOWPACK (black dashed), IMAU-FDM (yellow), and GSFC-FDM (red dashed) density bias profiles at the 69 sites not included in the IMAU-FDM and GSFC-FDM density calibrations (b). Error bars represent 95% confidence intervals on the bias means. Map of 69 observed density profiles not included in the IMAU-FDM and GSFC-FDM density calibrations (c).

regimes which are well sampled by observations. Alternatively, SNOWPACK, and other physics-based models, simulate density profiles which are not calibrated against observations and could therefore, potentially outperform semi-empirical models in climate regimes not well sampled in semi-empirical model calibrations. To test this hypothesis, we examine the 69 of 122 observed density profiles which were not included in the IMAU-FDM and GSFC-FDM calibrations, hereafter referred to as the 69 independent sites. We then compare the mean near-surface density biases of SNOWPACK, IMAU-FDM, and GSFC-FDM at these 69 independent sites against the same statistics calculated for all 122 sites (Section 3.6).

Figure 10 shows simulated density profiles along with observations and biases at the 69 sites not used for calibration. The SNOWPACK mean density bias for the 69 independent sites is 1.1 $\mathrm{kg\,m^{-3}}$, down from -9.7 $\mathrm{kg\,m^{-3}}$ for all 122 sites. The IMAU-FDM also exhibits a smaller absolute bias, increasing from -32.5 to -20.4 $\mathrm{kg\,m^{-3}}$ for the 69 and 122 sites, respectively. However, the SNOWPACK model demonstrates a consistent performance, and smaller absolute bias than IMAU-FDM, irrespective of including the calibration sites. For the GSFC-FDM model, the mean density bias increases throughout the near-surface from 15.5 to 22.3 $\mathrm{kg\,m^{-3}}$ for the 69 and 122 sites, respectively, indicating a degraded performance when calibration sites are excluded. These results show that the fact that physics-based firn models do not need specific calibration for the application, amd could yield an advantage over calibrated firn models, where the application for non-calibrated sites is not guaranteed to yield good performance.

### 3.8 SNOWPACK and semi-empirical models hierarchical complexity

Although all three models are presented on similar footing, it is important to contextualize their performance by noting the different original purposes for developing SNOWPACK, IMAU-FDM, and GSFC-FDM, as well as well as their different level of complexity in representing physical processes. The IMAU-FDM and GSFC-FDM are relatively simplified semi-empirical firn models designed to represent spatial and temporal evolution of firn density and surface height. On the other hand, SNOWPACK is a higher order complexity physics-based land-surface snow model originally intended for simulating snow microstructural properties relevant for avalanche formation in seasonally snow covered terrain. IMAU-FDM and GSFC-FDM are usually used to study the entire ice sheet firn column, which can be greater than $100 \, \mathrm{m}$ thick, while SNOWPACK is typically used to simulate seasonal snowpacks a few meters thick. Despite the much thicker simulated firn column in IMAU-FDM and GSFC-FDM ($40 - 120 \, \mathrm{m}$) when compared to our imposed restriction for SNOWPACK to the near-surface (i.e. depths $0 - 10 \, \mathrm{m}$), SNOWPACK is considerably more computationally expensive to run. It is therefore important to acknowledge the inherent trade off between process representation and cost, particularly for simulations which require long spinups or involve long time frames, e.g. paleoclimate applications or future climate scenarios.

Because SNOWPACK exhibits a lower mean snow and firn density bias throughout most of the near-surface compared to IMAU-FDM and GSFC-FDM, it is necessary to describe particular model distinctions which could explain their different behavior. First, SNOWPACK and IMAU-FDM rely on different meteorological forcing (MERRA-2 and RACMO2 2.3p2, respectively) which somewhat confounds their direct comparison. Next, in our implementation of SNOWPACK, new snowfall density is determined by hourly mean MERRA-2 weather conditions while in IMAU-FDM and GSFC-FDM, new snow density is determined by annual average meteorologic variables including accumulation rate, 10 m wind speed, and surface temperature. SNOWPACK therefore calculates a variable new snow density driven by both seasonal and synoptic scale variability in surface meteorology, while IMAU-FDM and GSFC-FDM do not resolve this process. Additionally, SNOWPACK allows for drifting snow compaction of previously fallen snow (Fig. 1), while IMAU-FDM and GSFC-FDM only account for the impact of wind on density by including this processes in the new snow density parameterization. This difference in process representation of new snow density and drifting snow compaction is reflected by SNOWPACK's larger variability in predicted surface density ($281 - 479 \, \mathrm{kg \, m^{-3}}$) compared to IMAU-FDM ($328 - 429 \, \mathrm{kg \, m^{-3}}$) and GSFC-FDM ($364 - 436 \, \mathrm{kg \, m^{-3}}$). By allowing for initial low density new snow accumulation and subsequent drifting snow compaction, SNOWPACK, more in line with observations ($272 - 507 \, \mathrm{kg \, m^{-3}}$), predicts a wider range of surface snow densities than IMAU-FDM and GSFC-FDM (Fig. 7, panels a, c, e). Finally, SNOWPACK's higher temporal resolution density output (daily) compared to IMAU-FDM (30 days) and GSFC-FDM (5 days), can resolve processes relevant for surface snow density which act on timescales of less than 30 and 5 days, respectively (e.g. snowfall and drifting snow compaction).

### 3.9 SNOWPACK simulated surface density variability

SNOWPACK predicts a wide range of simulated surface densities (mean of top meter), ranging from $230 - 684 \, \mathrm{kg \, m^{-3}}$ with a mean of $370 \, \mathrm{kg \, m^{-3}}$ and standard deviation of $64 \, \mathrm{kg \, m^{-3}}$ across 186 simulations (Fig. 11a). Furthermore, SNOWPACK

exhibits large seasonal surface density variability (mean magnitude = 53 $\mathrm{kg\,m^{-3}}$), when measured as the maximum difference between summer and winter surface density found in the period 1980 – 2017 (Fig. 11b). Seasonal surface density variability is primarily driven by seasonal cycles in temperature, accumulation rate, and compaction rate. However, superimposed onto these processes at even shorter timescales (e.g. hourly to weekly) are individual accumulation events and snow microstructural evolution (Fig. 11c). Because satellite altimetry relies on instantaneous measurements of surface height, rather than temporal means, firn models used to reliably convert changes in volume into mass must capture high frequency variability in firn density and surface height.

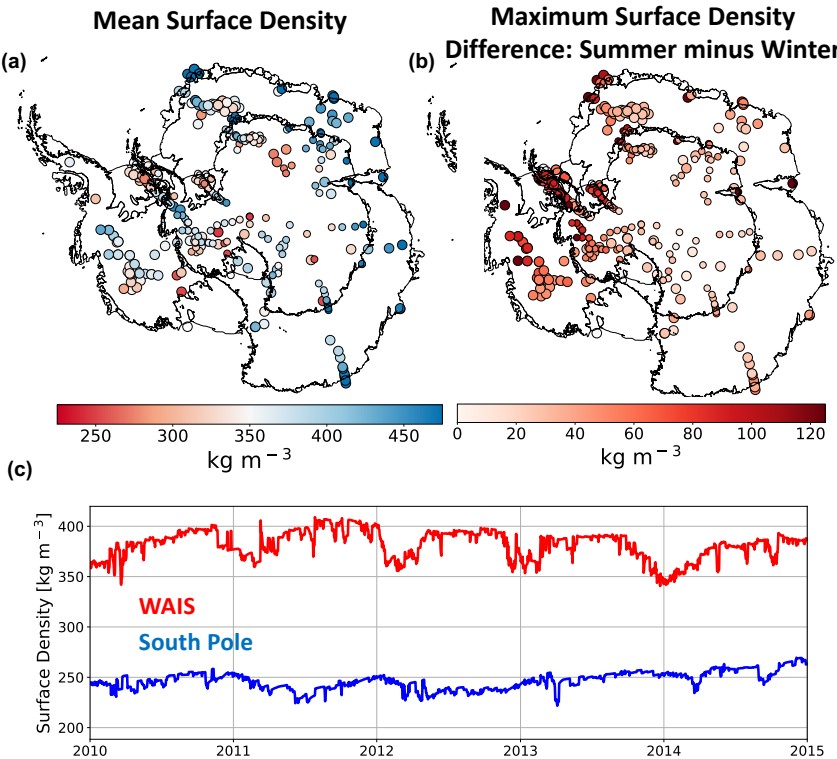

**Figure 11. SNOWPACK simulated surface snow density.** Map of 1980 – 2017 mean SNOWPACK surface density ($\mathrm{kg\,m^{-3}}$, a). Map of SNOWPACK simulated surface density variability, defined as the 1980 – 2017 maximum difference between summer and winter mean surface density ($\mathrm{kg\,m^{-3}}$, b). Time series of 2010 – 2015 SNOWPACK simulated average surface density ($\mathrm{kg\,m^{-3}}$) at South Pole (blue) and WAIS (red, c).

## 4   Conclusions

Accurate snow and firn density models are required to reliably determine ice sheet mass balance using satellite altimetry. However, firn densification models currently used in altimetry studies do not resolve observed temporal increases in surface

snow density under the influence of wind. In this study, we demonstrate improved simulation of Antarctic near-surface (depths $\leq 10$ m) snow and firn density upon implementation of a new drifting snow compaction routine into SNOWPACK, a detailed, physics-based land surface snow model. In particular, we show that when compared to two other semi-empirical firn densification models (IMAU-FDM and GSFC-FDM), SNOWPACK exhibits a lower mean snow and firn density bias throughout most of the near-surface at 122 observed density profiles across the Antarctic ice sheet. Despite this improvement, SNOWPACK generally underestimates density, with a mean surface density bias of -8.2 $\mathrm{kg\,m^{-3}}$ and mean near-surface biases ranging from -0.6 $\mathrm{kg\,m^{-3}}$ at $2-3$ m depth to -25.3 $\mathrm{kg\,m^{-3}}$ at $8-9$ m depth. Meanwhile, IMAU-FDM exhibits a mean surface density bias of -20.4 $\mathrm{kg\,m^{-3}}$ and mean near-surface biases between -20.4 $\mathrm{kg\,m^{-3}}$ at depths $0-1$ m to -40.1 $\mathrm{kg\,m^{-3}}$ at depths $8-9$ m while GSFC-FDM shows a mean surface density bias of 20.4 $\mathrm{kg\,m^{-3}}$ and mean near-surface biases from 8.5 $\mathrm{kg\,m^{-3}}$ at depths $8-9$ m to 23.7 $\mathrm{kg\,m^{-3}}$ at $2-3$ m. SNOWPACK exhibits a lower density bias than IMAU-FDM throughout the entire near-surface, however the GSFC-FDM shows a lower magnitude bias than SNOWPACK below 7 m depth and throughout the entire near-surface at high accumulation sites. Note that our analysis is limited to the top 10 m in order to focus on the most dynamic and variable part of the Antarctic firn layer. Because SNOWPACK is a physics-based model, extensive model tuning in order to fit observations is not required. By analyzing the simulations, excluding the sites used for calibration of the semi-eprical models, we found SNOWPACK to also have the lowest absolute bias. IMAU-FDM showed a lower bias when excluding the calibration sites, whereas GSFC-FDM showed larger bias. Because SNOWPACK outperforms the IMAU-FDM and GSFC-FDM at sites not included in calibration, SNOWPACK, compared to semi-empirical models, could possibly simulate firn density more accurately in regions without extensive observations or under future climate scenarios, where firn properties are expected to diverge from their current state.

*Code and data availability.* All software and code required to replicate this study is available open-access at https://github.com/EricKeenan/Keenan_et_al_2020_TC. SNOWPACK model source code can be accessed at https://github.com/snowpack-model/snowpack, while the precise version used in this study can be accessed at https://doi.org/10.5281/zenodo.3891846. MERRA-2 atmospheric reanalysis is available at https://gmao.gsfc.nasa.gov/reanalysis/MERRA-2/ and can be retrieved and processed using our workflow available at https://github.com/EricKeenan/download_MERRA2. SUMup density data are available at https://arcticdata.io/catalog/view/doi:10.18739/A26D5PB2S. Borehole 10 m depth temperature data are available from Michiel van den Broeke (m.r.vandenBroeke@uu.nl). AWS data are available from Carleen Reijmer (c.h.tijm-reijmer@uu.nl). IMAU-FDM data are available from Peter Kuipers Munneke (p.kuipersmunneke@uu.nl). GSFC-FDM data are available from Brooke Medley (brooke.c.medley@nasa.gov).

## Appendix A

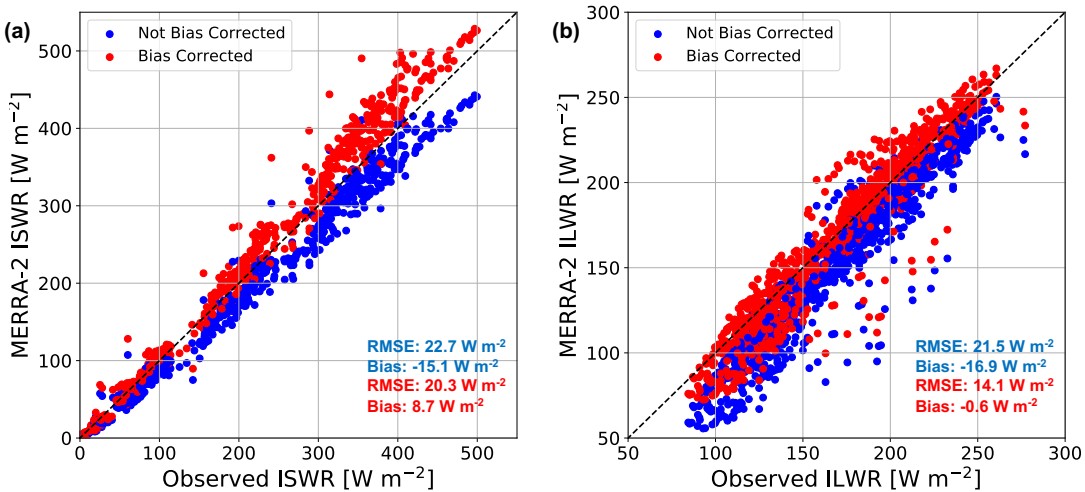

**Figure A1. MERRA-2 incoming shortwave and longwave radiation bias correction.** Comparison between monthly averaged AWS observed and MERRA-2 modeled incoming shortwave radiation (a) and longwave radiation (b) at nine automatic weather stations. Non-bias corrected MERRA-2 radiative fluxes are shown in blue while bias corrected fluxes are shown in red. Mean MERRA-2 root mean square error (RMSE, $W\,m^{-2}$) and bias ($W\,m^{-2}$) are shown in blue and red text for non bias corrected and bias corrected radiative fluxes, respectively. Dashed black line represents a one to one line.

*Author contributions.* All authors contributed to preparing this manuscript. E.K. carried out SNOWPACK simulations, performed the analysis, and produced the figures. N.W. implemented the new drifting snow compaction routine into SNOWPACK and contributed to the experimental design and analysis. M.D and J.T.M.L. contributed to the experimental design and analysis. B.M. contributed to the experimental design and analysis and provided GSFC-FDM output. P.K.M provided IMAU-FDM output. C.R. provided AWS data.

*Competing interests.* The authors declare that they have no conflict of interest.

*Acknowledgements.* E.K., N.W., J.T.M.L., and B.M. acknowledge support from the National Aeronautics and Space Administration (NASA), Grant 80NSSC18K0201 (ROSES-2016: studies with ICESat-2 and CryoSat-2). E.K., N.W., and J.T.M.L are also supported by BELSPO Research Contract, grant BR/165/A2:Mass2Ant. P.K.M. is funded by the Netherlands Earth System Science Centre (NESSC). C.R. acknowledges the support of the Dutch Polar program of the Dutch national research council NPP-NWO. This work utilized the RMACC Summit supercomputer, which is supported by the National Science Foundation (awards ACI-1532235 and ACI-1532236), the University of Colorado

Boulder, and Colorado State University. The Summit supercomputer is a joint effort of the University of Colorado Boulder and Colorado
State University. Data storage supported by the University of Colorado Boulder "PetaLibrary". The authors thank Lynn Montgomery for their
useful insight into the SUMup dataset and constructive comments on this manuscript.

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
