# Peer review of "Physics-based SNOWPACK model improves representation of near-surface Antarctic snow and firn density"

_The Cryosphere, 2020_

## Referee Comment (RC1) · Charles Amory (Referee) · 31 Jul 2020

General comments

The paper presents an evaluation of the 1D model SNOWPACK equipped with a drifting-snow compaction routine against a large set of observed density profiles and 10 m depth temperatures scattered over Antarctica. I really enjoyed reading the paper which is nicely formulated and timely as the role of drifting-snow compaction is a currently lacking process in most of snow transport models that parameterize surface density following semi-empirical formulations designed to reproduce observed density already resulting from post-depositional processes. This is moreover an important topic for estimations of ice-sheet contribution to sea level rise since density is needed

for converting altimetry-derived ice volumes to mass. I think however that the paper could benefit from substantial improvements before publication.

Major comments

Important model information is generally lacking notably about initialization conditions, number of snow/ice layers, vertical discretization, treatment of concurrent deposition of snowfall during drifting snow, layer aggregation, and more generally on the actual influence of the inclusion of the drifting-snow compaction process on the model density products. The evaluation of forcing wind speeds taken from MERRA-2 reanalysis, which is the driving force behind drifting snow and related compaction, could be more exhaustive and adapted to more relevant characteristic time scales of the process studied. Drifting-snow compaction is only able to densifies snow at the surface (« 1m, which is by the way a coarse definition of the real surface layer affected by drifting snow in natural environments), so the only process by which drift-induced density anomalies can be buried is by accumulation. It's not clear to me what is the added-value of including drifting-snow compaction in the representation of density profiles. In many places the paper focuses on discussing the representation of density at depths at which drifting-snow compaction is not active, so the contribution of overburden pressure alone to the good agreement with observations and more generally to density profiles cannot be assessed. This could however be easily done by running the model without the drifting-snow compaction routine, as suggested in my comments. By doing so, the interesting question of "To which depth drifting-snow compaction impacts density profiles?" could be answered and help to shape our understanding on that poorly documented process while improving the scientific significance of the paper.

Minor comments

- L25, "drifting snow": Many authors have referred to drifting (or blowing) snow for describing different processes (saltation, combined or not with suspension, wind-driven snow transport > 2 m and/or < 2 m, local erosion combined or not with horizontal

advection, etc..) leading to a potential confusion of the actual meaning of this term when not properly defined. Could you describe which specific(s) process(es) you refer to?

- L26, "wind-driven compaction": please elaborate a bit on the physical mechanisms behind drifting-snow compaction (mainly through rounding and fragmentation) as it is a key element of the paper.

- L51: I don't understand where in the paper SNOWPACK is applied to the 9 AWS locations, and what would be the objective of doing so if only meteorological observations are available there.

- L69, "both ice sheets": I suggest to add "both the Greenland and Antarctic ice sheets".

- L70-71: the SEB is not a process. Could you please reformulate?

- L92: Have you investigated the sensitivity of your results to the choice of the roughness length or the neutral atmosphere assumption? 2 mm is a rough value that do not necessarily fit with observations all over the AIS (see for instance Amory et al., 2017; Vignon et al., 2016), and the Antarctic ABL is more generally statically stable in the ice-sheet interior, requiring the use of stability correction functions which can be a significant source of uncertainty in the computation of u* (Vignon et al., 2016).

- L99: If Q is a saltation mass transport rate, then phi is simply a saltation (drifting snow) mass flux.

- Fig. 1, and elsewhere in the paper : "Drifting snow erosion" and "Drifting snow redeposition" sound like a pleonasm, as drifting snow is the very process by which surface snow is eroded and redeposited. Consider simply using erosion and redeposition.

- L102, "are distributed before erosion": Distribution involves erosion. Not sure what do you mean here.

- L107: Why just not saying here that phi and Q only accounts for saltation? Due to

the one-dimensional approach, a missing aspect is horizontal advection of snow which can significantly contribute to the local saltation mass flux. Even though the objective of the paper is not to parameterize explicitly (tri-dimensional effects of) snow transport, this could lead to overestimation of drifting-snow compaction if all the saltation mass flux is attributed to local erosion.

- L109: "fresh surface snow" can be confusing since this equation has been developed to account for deposition of snow that has been transported by the wind only (see Groot Zwaaftink et al. 2013, p337), while "fresh snow" could refer to snow originating from clouds and that has reached the ground for the first time. It is maybe preferable to stay in line with the semantics of Groot Zwaaftink et al. (2013) and just remove "fresh".

- L112-113: Here the assumption is made that all the eroded snow is redeposited. How are the internal layers of the snowpack affected by deposition of new snow layers of different densities? How is the mixing with snowfall treated to compute the density of the surface layer in case of concurrent snowfall?

- L119-121: I don't understand why the release latency and update should constitute an argument since you focus on a past period (1980-2017) over which RCM outputs are already available (see for instance van Wessem et al., 2018, Agosta et al., 2019).

- L122-124: The good agreement reported by Gossart et al. (2019) is demonstrated from mean values of mean values and thus remains valid for discussing the mean climatology of Antarctica, But that reference could hardly be used to demonstrate the ability of MERRA-2 to represent climate variables at the 6-hourly time scale or at least at the characteristic time scale over which ephemeral processes such as drifting-snow compaction is active.

- L125: Could you justify why do you prefer the monthly scale when evaluation at the daily scale, or even less (SNOWPACK is forced at hourly intervals), could be similarly performed to better highlight MERRA-2 ability at reproducing the meteorology, moreover required as input in Eq. (1)? Strong wind events, during which most of

drifting-snow compaction occurs, are completely smoothed out at the monthly scale.

- L130: The performance of MERRA-2 at reproducing the Antarctic near-surface mete-orology (i.e., « month) is still poorly known. While this is certainly the subject of another study and lies beyond the scope of the paper, still you have all the materials required to do it, and this could be a real added value to your work while reinforcing the evaluation. This is also, again, more consistent with the time scale of drifting-snow compaction. At least could you give more statistics, i.e. RMSE and r2, which are better indicators (when combined together) than just a mean bias, to support your assertion. Moreover, I get that these 9 AWSs are not assimilated in MERRA-2 so they are all good and in-dependent evaluation products. But why so few AWSs when many others (>200, see Mottram et al., 2020) are available through other public sources and not necessarily as-similated in MERRA-2 (see https://gmao.gsfc.nasa.gov/pubs/docs/McCarty885.pdf)? Antarctic is vast and diverse. Most of the AWSs used here are located in DML and is a rather small sample of the climate conditions encountered across the continent. Could you add more AWSs to your analysis, or at least discuss the representativeness of these locations regarding the Antarctic climate conditions, also given that the eval-uation using boreholes T and density profiles is mostly done at locations significantly away from the AWSs?

- L133: Do you rather mean -15.1 W/2, so the applied correction correspond to the mean bias as done for ILWR? If not, where does this value come from?

- L139: Important information are missing here, such as the initialization, the number of ice/snow layers, the vertical resolution of SNOWPACK and aggregation of new snow layers. You must elaborate on this.

- L171: Defining the surface as the 1st meter is quite coarse regarding the actual thick-ness of the layer affected by surface post-depositional processes. For instance, Groot Zwaaftink et al. (2013) consider the first 10 cm. Specifying the timing at which you compare model with observations can be of significant importance for these surface

layers (« 1m) depending on the recent occurrence of melt, snowfall and drifting-snow events, more importantly given that the interest of implementing a drifting-snow compaction routine partly relies on improving representation of density at the surface at the time of drifting-snow events. I would expect more details about the comparison methodology. Do you compare observed profiles with mean modelled profiles ? for which period ?

- L189, "almost perfect" : Sounds a bit too emphatic. I would advise to remain neutral when describing your results. A bias value alone, even low, is not a self-consistent argument to speak of almost perfect agreement when RMSE still amounts to 2.36 ° C (indicating individual bias values of several degrees in some locations).

- L199: Did you follow the same spin-up procedure for your sensitivity analysis?

- L200: Why did you choose these specific locations? Please justify and give coordinates.

- L211-212: This is another strong argument for exploring the sensitivity of density to the derivation method for u* (z0, stability correction function) as well as for evaluating MERRA-2 wind speeds at shorter intervals.

- L214: This may be because wind maxima which control drifting-snow occurrence and thus drifting-snow compaction, are smoothed out at such a low (monthly) temporal resolution. Does this stay true if you perform a statistical evaluation of wind speed at higher temporal resolution?

- L215-216: Again, then knowing the sensitivity of your results to the derivation method of u* would be particularly interesting.

- L227-229: I couldn't agree more. This is another argument in favor of an evaluation of wind speed at higher temporal resolution. Another explanation to this result might be that drifting-snow compaction is mainly active over layers thinner than 1 m. What would the correlation become by decreasing the size of the surface layer consider here

while working over shorter time scales?

- L280-281: I'm wondering to which depth drift-induced compaction exerts an influence on the mean density profile, given the fact that only the surface layer receives momentum from the atmosphere and is likely affected by drifting snow. This is a very interesting question, I don't have the answer and your work is among the first to focus on this aspect. But you could give an element of response by running the model without the drift compaction routine and see how it affects density profiles (and which layers are most impacted) according to SNOWPACK pre-existing physics by comparing it with the run including drifting-snow compaction.

- L301: No new results in this section, but these are good elements of discussion. This should be entirely part of a Discussion section, or mixed with the Conclusion re-entitled Discussion and Conclusion.

- L317: "new snow density": specify if new=deposited (snowfall) or redeposited (drifting snow). Maybe consider staying in line with Groot Zwaaftink et al. 2013 in which "new snow" is defined as redeposited.

- L331: This section is out of the scope, of limited interest and with no scientific results. Everything here could be moved to the conclusion and resumed in one sentence informing on the availability of SNOWPACK products for other possible applications.

- L350-352: You need a comparison between runs with and without the compaction routine to clearly highlight improvements and state that drifting-snow compaction is the process behind this. Besides, it would be a very interesting results that would strengthen the scientific contribution of the paper.

Technical corrections

- L5, "wind-driven, drifting snow": either wind-driven or drifting snow, but a combination of both is redundant.

- L179, "absent of very rare": Please add a reference.

- L224, "perhaps surprisingly": Avoiding subjective wording is strongly recommended.

- L263: remove "at depth".

- L273, "off": on?

- L325: correct "comapred".

- L336,"valide": "evaluate" is more appropriate.

Agosta, C., Amory, C., Kittel, C., Orsi, A., Favier, V., Gallée, H., van den Broeke, M. R., Lenaerts, J. T. M., vanWessem, J. M., van de Berg, W. J., and Fettweis, X.: Estimation of the Antarctic surface mass balance using the regional climate model MAR (1979–2015) and identification of dominant processes, The Cryosphere, 13, 281–296, https://doi.org/10.5194/tc-13-281-2019, 2019.

Amory, C., Gallée, H., Naaim-Bouvet, F., Favier, V., Vignon, E., Picard, G., Trouvilliez, A., Piard, L., Genthon, C., and Bellot, H.: Seasonal Variations in Drag Coefficient over a Sastrugi- Covered Snowfield in Coastal East Antarctica, Bound.-Lay. Meteorol., 164, 107–133, https://doi.org/10.1007/s10546-017-0242-5, 2017.

Gossart, A., Helsen, S., Lenaerts, J. T. M., Broucke, S. V., van Lipzig, N. P. M., and Souverijns, N.: An Evaluation of Surface Climatology in State-of-the-Art Reanalyses over the Antarctic Ice Sheet, Journal of Climate, 32, 6899–6915, https://doi.org/10.1175/JCLI-D-19-0030.1, 2019.

Groot Zwaaftink, C. D., Cagnati, A., Crepaz, A., Fierz, C., Macelloni, G., Valt, M., and Lehning, M.: Event-driven deposition of snow on the Antarctic Plateau: analyzing field measurements with SNOWPACK, The Cryosphere, 7, 333–347, https://www.the-cryosphere.net/7/333/2013/, 2013.

Vignon, E., Genthon, C., Barral, H., Amory, C., Picard, G., Gallée, H., Casasanta, G., and Argentini, S.: Momentum and heat-flux parametrization at Dome C, Antarctica: a sensitivity study, Bound.-Lay. Meteorol., 162, 341–367,

https://doi.org/10.1007/s10546-016-0192-3, 2016.

van Wessem, J. M., van de Berg, W. J., Noël, B. P. Y., van Meijgaard, E., Amory, C., Birnbaum, G., Jakobs, C. L., KruÌĹger, K., Lenaerts, J. T. M., Lhermitte, S., Ligtenberg, S. R. M., Medley, B., Reijmer, C. H., van Tricht, K., Trusel, L. D., van Ulft, L. H., Wouters, B., Wuite, J., and van den Broeke, M. R.: Modelling the climate and surface mass balance of polar ice sheets using RACMO2 –Part 2: Antarctica (1979–2016), The Cryosphere, 12, 1479–1498, https://doi.org/10.5194/tc-12-1479-2018, 2018.

---

## Referee Comment (RC2) · Anonymous Referee #2 · 25 Sep 2020

**Review: Physics-based modeling of Antarctic snow and firn density**

Eric Keenan, Nander Wever, Marissa Dattler, Jan T. M. Lenaerts, Brooke Medley, Peter Kuipers Munneke, and Carleen Reijmer

**Summary**
The authors apply the physically-based SNOWPACK snow model over the Antarctic ice sheet in order to simulate snow and firn densification. The SNOWPACK simulation is compared with in situ measurements and two other firn densification models. The authors find that biases in SNOWPACK are generally lower than in the two other empirically-based models, especially for locations where observations have not been used to calibrate the semi-empirical models. They suggest that in future projections of Antarctic firn densification, SNOWPACK would produce more reliable results, because of more detailed representation of physical processes, compared with simpler semi-empirical models.

**General Comments**
In general, the manuscript is well written and well organized. The topic is an important and relevant one, especially given the recent launch of the ICESat-2 altimetry satellite. The scheme introduced is more sophisticated than that of other models applied over the Antarctic ice sheet, and the paper therefore represents an important advance on other recent studies. The paper clearly confirms that the SNOWPACK model is capable of realistically simulating near-surface density over the Antarctic ice sheet, and has advantages over other models in that it is more detailed in its representation of physical processes governing snow evolution. I do have some concerns regarding the interpretation of results, however, as noted below. I feel that overall these revisions do not require major changes to the paper, but should be addressed before the paper can be published.

(1) Given that the uncertainty ranges of the density simulated by the different models overlap to some degree, it is not completely clear whether there is a statistically significant difference between them, or between the models and the observations at different levels. The authors should test whether this is the case.

(2) The authors should be careful to note some of the limitations of the current implementation (e.g. the validation is over the top 10 m, not the entire firn column; and the SNOWPACK bias is larger below 6 m depth) particularly in the abstract and conclusions sections.

(3) The available evidence doesn't seem to necessarily support the argument that biases are substantially larger in the semi-empirical models at locations that were not used to calibrate those models. The authors should clarify whether this is indeed the case and revise the text accordingly. It would be interesting to include both the GSFC-FDM and IMAU-FDM in this comparison if possible. Further specific comments are provided below.

**Specific Comments**

1. **Title:** I would suggest adding SNOWPACK to the title, and mentioning the near-surface e.g. "Physics-based modeling of near-surface Antarctic snow and firn density with the SNOWPACK model". I would argue that the other models utilized here are also physically-based, they just employ simpler parameterizations for the process of firn densification.
2. **Lines 1-11:** In general, some quantitative evidence should be provided here. Some of the limitations of SNOWPACK applied over Antarctica should be discussed, for example the larger bias for higher accumulation areas and the larger biases deeper in the snowpack, as well as the fact that this approach focuses on the near-surface, not the full firn column.
3. **Lines 7-8:** It would be best to quantify the magnitude of the biases here.
4. **Line 9:** It isn't entirely clear from this sentence that this is one of the findings of the study; it would be best to provide some quantitative results here. Also I believe the performance does degrade somewhat at these sites, just not as much as for the semi-empirical models?
5. **Line 17:** It would be informative to mention other methods of estimating mass balance (e.g. gravity measurements, e.g. Velicogna et al., 2020; or the input output method, e.g. Rignot et al., 2019).

   Velicogna, I., Mohajerani, Y., A, G., Landerer, F., Mouginot, J., Noël, B., Rignot, E., Sutterley, T., van den Broeke, M. R., van Wessem, M., and Wiese, D. (2020) Continuity of ice sheet mass loss in Greenland and Antarctica from the GRACE and GRACE Follow-On missions. *Geophysical Research Letters* 47, e2020GL087291.

   Rignot, E., Mouginot, J., Scheuchl, B., van den Broeke, M., van Wessem, M. J., and Morlighem, M. (2019) Four decades of Antarctic Ice Sheet mass balance from 1979-2017. *Proceedings of the National Academy of Sciences*, 116, 4, 1095-1103.

6. **Line 38:** What is meant by "all local and temporal density variability"? No model can capture "all" variability. Please clarify.
7. **Line 41:** These models do employ "physical principles"; they are not entirely empirical. Suggest simply removing the phrase "rather than physical principles".
8. **Lines 50-53:** Describe how the model is forced, briefly.
9. **Line 50:** Instead of "we apply", do you mean "we compare results from">
10. **Line 61:** SNOWPACK also seems to include parameterizations that are empirically based. Perhaps mention explicitly how SNOWPACK is different from the other models mentioned in earlier sections.
11. **Line 75:** Perhaps change "new drifting snow compaction routine" to "new snow compaction routine", as drifting snow is just a component of the routine.
12. **Line 80:** Can the authors briefly note how this parameterization is derived?

13. **Lines 87-88:** How much do these parameters change the comparison with observed profiles. Provide some additional details either in the main manuscript or a supplemental section.
14. **Line 90**: Briefly explain the physical meaning of the "threshold friction velocity".
15. **Line 124**: Is this a bias over the entire Antarctic ice sheet? Are there spatial variations in the bias?
16. **Lines 130-131:** Is there a reference for these statements?
17. **Line 133:** Why use 19.4% and not 15.1 W m$^{-2}$ ?
18. **Line 134:** Why is there still a bias after the bias is removed?
19. **Lines 154-170:** It would be helpful here to describe these two models in a bit more detail, in particular to highlight how they differ from SNOWPACK in terms of key physical processes (e.g. compaction), as the model differences are important to the conclusions of the study.
20. **Line 166:** Explain the meaning of "replay".
21. **Line 194:** Suggest changing "reduction in both RMSE…" to "statistically significant reduction in both RMSE…"
22. **Line 196:** This section could potentially be moved to later in the manuscript. It might logically follow the section on comparison with observations.
23. **Line 200:** Clarify why these two stations were chosen.
24. **Line 221:** It is a bit unclear what is meant by "we tested for explanatory variables". Please clarify.
25. **Lines 245-247:** It might be useful to have a table here for the bias and RMSE for different models above and below 400 kg m$^{-3}$.
26. **Line 253:** This sentence is confusing. Suggest revising to read something like: "Additionally, we cannot rule out the possibility of larger errors in the observational data for densities above 400 kg m$^{-3}$."
27. **Line 256:** The SUMup dataset does include information on measurement methods. It might be interesting to see if dividing by measurement method changes these biases in any way.
28. **Lines 259-271, Fig. 6:** Can the authors note whether the differences are statistically significant? It might also be useful to provide an uncertainty range on the biases. Also, at first glance at it appears that all the model simulation uncertainty ranges overlap in Fig. 6, but this is not the case. Perhaps the figure can be modified slightly to make this clearer, e.g. changing the transparency for different models or changing the colors. (Not sure how easy this would be).
29. **Lines 293-294:** It seems this would not be difficult to find out? It would also be interesting to see the IMAU-FDM results.
30. **Lines 296-297:** From Fig. 8, it actually looks like there is a larger change in the SNOWPACK density bias (at least at different levels). The numbers here do not seem to match with the figure. Please clarify.
31. **Lines 335-336:** This portion is interesting but seems disconnected from the rest of the manuscript. Perhaps these temporal variations could be placed in the context of temporal variations from in situ data. Are there any locations

where a timeseries of measurements is available that could be compared with the SNOWPACK runs?

32. **Line 342:** Without validation of the temporal variability of the in situ measurements, I'm not sure the model results would qualify as "evidence". Please revise.

33. **Lines 360-364:** I'm not sure these statement is completely supported by the results. For example, SNOWPACK seems to show a larger bias at higher accumulation locations, and the SNOWPACK and the GSFC-FDM both seem to show a positive bias in locations that were not used to constrain GSFC-FDM between 0 and 6 m in depth. In general, however, I would agree that including a more physically realistic simulation of snowpack processes should produce a better projection of future conditions. Perhaps revise this statement to note that this is likely the case, but not entirely certain.

**Technical Corrections**

1. **Line 14:** Change "with an increasing…" to "at an increasing…"
2. **Line 91:** Change "from MERRA-2" to "from the MERRA-2".
3. **Fig. 2 caption:** Change "SNOWPACK simulations" to "SNOWPACK simulation locations" for clarity. Note that the borehole depths are 10 m below the surface for clarity.
4. **Line 172:** Change "as depths" to "the average density between depths of"
5. **Line 185:** Change "of average" to "of the average".
6. **Line 189:** Suggest changing "almost perfect" to "an excellent".
7. **Line 198:** Change "is bias-corrected MERRA-2" to "in bias-corrected MERRA-2".
8. **Line 215:** Change "windspeed represents" to "density variations due to uncertainty in windspeed represent".
9. **Line 219:** Change "observations" to "observed density values".
10. **Lines 259-260:** This sentence is quite wordy. Suggest revising, e.g. "In a comparison at 122 observed density profiles, SNOWPACK exhibits a lower bias compared to IMAU-FDM for the entire near-surface, and a lower bias compared to GSFC FDM between from 0 to 7 m depth (Fig. 6)".
11. **Lines 273-274:** Suggest changing to read "low SMB categories by applying a threshold of 200 kg m$^{-2}$ yr$^{-1}$ to MERRA-2 mean annual SMB (Fig. 7)."
12. **Line 276:** Change "reduced" to "lower" or "smaller"
13. **Line 277:** Again change "reduced" to "lower" or "smaller".
14. **Caption, Fig.** 7: Change "GSFM-FDM" to "GSFC-FDM". Change "a MERRA-2 1980-2017 mean annual SMB threshold of 200 kg m$^{-2}$ yr$^{-1}$" to "a 200 kg m$^{-2}$ yr$^{-1}$ threshold applied to MERRA-2 1980-2017 mean annual SMB."
15. **Lines 303-304:** Change "as well as their different level of process representation complexity" to "as well as their different level of complexity in representing physical processes."
16. **Line 325:** Change "compared" to "compared".
17. **Line 373:** The heading for Appendix A is out of place.

---

## Author Comment (AC1) · 23 Nov 2020

First of all, thank you very much to editor Xavier Fettweis, first reviewer Charles Amory, and the anonymous second reviewer. We greatly appreciate both the positive feedback as well as constructive criticism from all. In order to address everyone's comments, we have responded individually below in blue. On behalf of all coauthors, we once again thank you for your time and effort.

On behalf all authors,

Eric Keenan

**Editor Review - Xavier Fettweis**

The manuscript presents robust and well discussed improvements in SNOWMODEL. The sensitivity to the forcing is particularly relevant. While it is a model development paper (fully fitting with Geoscientific Model Development (GMD)), it is within the scope and required initial quality level of TC. Therefore, I am happy to send it out for peer-review.

However, I'm bit afraid that the scientific relevance/conclusion of this paper remains a bit weak for TC (vs GMD) and that this weakness could be highlighted by the reviewers. Therefore, I suggest you to discuss more in depth the interest of your model with respect to the other ones for satellite altimetry.

Thank you very much to editor Xavier Fettweis for taking the time to consider our manuscript, as well as providing insightful comments. We have responded to each comment individually below.

I think that showing 2D maps of the surface (top 1m) density of each model as well as their time variability (standard deviation) will be interesting.

We believe the primary scientific interest of this paper is that the SNOWPACK model, following introduction of a new drifting snow compaction scheme, is able to reliably simulate Antarctic near-surface snow/firn density, and may outperform two existing models in the representation of near-surface firn density. To provide additional examples of SNOWPACK simulated densities, we have, along the lines of your suggestion, added maps of SNOWPACK simulated surface density and seasonal surface density variability at 186 sites across the Antarctic ice sheet (Figure 11). Due to present computational constraints, we have restricted these maps to previously completed SNOWPACK simulations as opposed to every MERRA-2 Antarctic grid cell.

In Fig 9, you suggest a significant time variability which could impact a lot the interpretation of the ice cores and suggests even that we need to take into account the time when the satellite altimetry measurements are made. Idea for the comparison with the ice cores (Fig5), I assume that you use the mean density over 1980-2017. How, by taking into account the time variability, this comparison could be impacted? For example by taking the summer or winter mean instead of the annual mean? In brief, what is the interest of better representing the time variability of snow density?

In Figure 7 (was Figure 5 in the first draft), SNOWPACK surface densities are, in contrast to 1980 - 2017 means, calculated from the closest daily model output to the

observation collection date declared in the SUMup dataset. We have clarified this in section 2.7.

According to SNOWPACK, surface density variability is significant. The maximum 1980 - 2017 surface density difference between summer and winter surface density averages 52.7 kg m$^{-3}$ (Figure 11) . We have elaborated on this interest in the introduction section:

"Semi-empirical densification models successfully capture broad regional variability in firn characteristics (van den Broeke, 2008; Ligtenberg et. al., 2008). However, due to their limited complexity, as measured by the inclusion of ephemeral processes such as drifting snow compaction, they cannot capture high frequency (i.e. hourly) temporal density variability at the snow surface, which is known to exist from field observations (e.g., Sommer at al. (2018), and may therefore provide inaccurate instantaneous density estimates which are ultimately used in satellite altimetry volume to mass conversions."

All of those are just suggestions to improve the scientific relevance of your paper. But if you judge that these are not useful or if you prefer to wait the comments from the reviewers, I'm OK to start the review round now with the present version of your manuscript as I have nothing to blame about the format and layout of your text, the legends and figures.

Thank you very much for the suggestions. We have critically evaluated them, and revised our manuscript accordingly.

Best regards,
Xavier F.

General comments

The paper presents an evaluation of the 1D model SNOWPACK equipped with a drifting-snow compaction routine against a large set of observed density profiles and 10 m depth temperatures scattered over Antarctica. I really enjoyed reading the paper which is nicely formulated and timely as the role of drifting-snow compaction is a currently lacking process in most of snow transport models that parameterize surface density following semi-empirical formulations designed to reproduce observed density already resulting from post-depositional processes. This is moreover an important topic for estimations of ice-sheet contribution to sea level rise since density is needed for converting altimetry-derived ice volumes to mass. I think however that the paper could benefit from substantial improvements before publication.

Many thanks to Reviewer Charles Amory for the positive notes on our manuscript as well as the specific points on how we can improve the paper. In recognition of the time the reviewer put into evaluating our manuscript, please find our detailed response below.

Major comments

Important model information is generally lacking notably about initialization conditions, number of snow/ice layers, vertical discretization, treatment of concurrent deposition of snowfall during drifting snow, layer aggregation, and more generally on the actual influence of the inclusion of the drifting-snow compaction process on the model density products. The evaluation of forcing wind speeds taken from MERRA-2 reanalysis, which is the driving force behind drifting snow and related compaction, could be more exhaustive and adapted to more relevant characteristic time scales of the process studied. Drifting-snow compaction is only able to densifies snow at the surface (« 1m, which is by the way a coarse definition of the real surface layer affected by drifting snow in natural environments), so the only process by which drift-induced density anomalies can be buried is by accumulation. It's not clear to me what is the added value of including drifting-snow compaction in the representation of density profiles. In many places the paper focuses on discussing the representation of density at depths at which drifting-snow compaction is not active, so the contribution of overburden pressure alone to the good agreement with observations and more generally to density profiles cannot be assessed. This could however be easily done by running the model without the drifting-snow compaction routine, as suggested in my comments. By doing so, the interesting question of "To which depth drifting-snow compaction impacts density profiles?" could be answered and help to shape our understanding on that poorly documented process while improving the scientific significance of the paper.

Thank you very much for raising these excellent points. As each is listed individually below, please find our detailed and specific comments below as well.

Minor comments

L25, "drifting snow": Many authors have referred to drifting (or blowing) snow for describing different processes (saltation, combined or not with suspension, wind-driven snow transport > 2 m and/or < 2 m, local erosion combined or not with horizontal advection, etc..) leading to a potential confusion of the actual meaning of this term when not properly defined. Could you describe which specific(s) process(es) you refer to?

Given our one dimensional modeling approach, we are unable to resolve three dimensional wind fields needed to realistically describe suspension of drifting snow. Therefore, we define drifting snow to represent saltation processes within the lowermost 2 m of the atmosphere. We have clarified this in the following sentence:

"In particular, surface snow and firn density are known to be strongly impacted by wind driven compaction, a process hereafter referred to as drifting snow compaction, whereby mobilized snow particles in the saltation layer (within the lowermost 2 m of the atmosphere) break apart upon collision with the snow surface. This process results in fragmented and rounded grains which pack together more efficiently, resulting in increased density (Vionnet et. al., 2012)."

L26, "wind-driven compaction": please elaborate a bit on the physical mechanisms behind drifting-snow compaction (mainly through rounding and fragmentation) as it is a key element of the paper.

This is a good point! We have now elaborated on the physical drivers behind drifting snow compaction - see added text in response to previous point.

L51: I don't understand where in the paper SNOWPACK is applied to the 9 AWS locations, and what would be the objective of doing so if only meteorological observations are available there.

Good point, this is something we overlooked. Indeed in the discussion paper the SNOWPACK simulations at the 9 AWS were only used to evaluate atmospheric forcing, rather than simulated snow properties. However in the revised paper we now present density results from these simulations in Figure 11, therefore we will leave things as they are.

L69, "both ice sheets": I suggest to add "both the Greenland and Antarctic ice sheets".

We have implemented this suggestion, thank you!

L70-71: the SEB is not a process. Could you please reformulate?

For clarity we have revised this sentence to the following:

"These studies have shown that SNOWPACK is capable of capturing important processes in the ice sheet firn layer, namely accumulation in windy environments, surface meltwater production, and subsequent liquid water retention in the firn."

L92: Have you investigated the sensitivity of your results to the choice of the roughness length or the neutral atmosphere assumption? 2 mm is a rough value that do not necessarily fit with observations all over the AIS (see for instance Amory et al., 2017; Vignon et al., 2016), and the Antarctic ABL is more generally statically stable in the ice-sheet interior, requiring the use of stability correction functions which can be a significant source of uncertainty in the computation of u* (Vignon et al., 2016).

There are several good points here, thank you for raising them.

We acknowledge that by prescribing a constant roughness length in both space and time, we ignore the substantial variations in roughness length on ice sheets as reported in observations (e.g. Amory et al., 2017; Vignon et al., 2016; Smeets and Van den Broeke, 2008). Since the complex interaction between surface roughness, turbulence, and drifting snow (e.g. Bintanja, 1998) is poorly constrained, and requires a highly detailed turbulence model, we have opted for the practical approach of employing a constant roughness length in SNOWPACK. To provide some insight into how this simplification affects our results, we note that in our modeling framework, the 10 m MERRA-2 wind speed is fixed. Therefore, by roughnening (smoothing) the snow surface, the friction velocity must increase (decrease), thus leading to both an increase (decrease) in the drifting snow frequency as well as saltation mass flux.

In terms of stability corrections, SNOWPACK relies on a logarithmic wind profile that is corrected for atmospheric stability (Michlmayr et. al., 2008). We will clarify this in the revised manuscript. Testing the sensitivity of our results to choice of stability corrections would be an interesting avenue for future research, but we would argue that this is beyond the scope of this paper.

L99: If Q is a saltation mass transport rate, then phi is simply a saltation (drifting snow) mass flux.

Good point, we have redefined phi as the saltation mass flux.

Fig. 1, and elsewhere in the paper : "Drifting snow erosion" and "Drifting snow rede-position" sound like a pleonasm, as drifting snow is the very process by which surface snow is eroded and redeposited. Consider simply using erosion and redeposition.

We have incorporated your suggestion into Figure 1 and have updated it to "erosion" and "redeposition" throughout the rest of the paper.

L102, "are distributed before erosion": Distribution involves erosion. Not sure what you mean here.

We have reformulated to the following: "L can be interpreted as a fetch length and characteristic horizontal length scale over which the originally upwind and now mobilized snow particles, which make up the saltation mass flux Φ, have been eroded from the snow surface".

L107: Why just not saying here that phi and Q only accounts for saltation? Due to the one-dimensional approach, a missing aspect is horizontal advection of snow which can significantly contribute to the local saltation mass flux. Even though the objective of the paper is not to parameterize explicitly (tri-dimensional effects of) snow transport, this could lead to overestimation of drifting-snow compaction if all the saltation mass flux is attributed to local erosion.

Good point. We agree that our one-dimensional model is not designed to resolve horizontal advection of suspended drifting snow. To make this more clear, we have edited the paragraph to include the following sentence, "Note that as suspension of drifting snow cannot be properly resolved by a one-dimensional snow model (Lehning and Fierz, 2008), the saltation mass transport rate Q and subsequent saltation mass flux Φ may underestimate the total mass flux in the saltation and suspension layers."

L109: "fresh surface snow" can be confusing since this equation has been developed to account for deposition of snow that has been transported by the wind only (see Groot Zwaaftink et al. 2013, p337), while "fresh snow" could refer to snow originating from clouds and that has reached the ground for the first time. It is maybe preferable to stay in line with the semantics of Groot Zwaaftink et al. (2013) and just remove "fresh".

Agreed! Thank you for pointing out this potentially confusing language. We have removed "fresh".

L112-113: Here the assumption is made that all the eroded snow is redeposited. How are the internal layers of the snowpack affected by deposition of new snow layers of different densities? How is the mixing with snowfall treated to compute the density of the surface layer in case of concurrent snowfall?

To address these questions we have added a new paragraph at the beginning of section 2.3.

"In our scheme, new snow layers are added on top of the modeled snow column when precipitation is present in the atmospheric forcing, in steps of 2 cm. Layers are initialized with a density given by Eq. 1 when they originate from precipitation. Layers originating from drifting snow are initialized with a density given by Eq. 4. Initial grain size for all newly added layers is 0.2 mm (Groot Zwaaftink et. al., 2013). There are two sets of microstructural properties for grain shape (dendricity and sphericity), for high and low wind speed, respectively (Groot Zwaaftink et. al., 2013). Note that precipitation is treated before assessing snow erosion, such that low density snow from precipitation can erode immediately when conditions allow. To reduce computation costs, a sophisticated snow layer merging scheme merges layers with very low ice content due to sublimation or melt and layers with similar properties (density, water content, grain size, and grain shape parameters). The criteria for layer merging are relaxed with depth, to allow for more aggressive layer merging with depth. At 10 m depth, typical layer spacing is around 10 - 20 cm. Near the surface, layers can be split to maintain a vertical resolution of a few cm near the surface, to be able to numerically represent steep temperature gradients."

L119-121: I don't understand why the release latency and update should constitute an argument since you focus on a past period (1980-2017) over which RCM outputs are already available (see for instance van Wessem et al., 2018, Agosta et al., 2019).

Low release latency is advantageous for timely estimation of snow properties, for example when interpreting satellite imagery and altimetry, or determining the status of field assets (e.g. weather stations). Indeed in this study we focus on 1980-2017, however we plan to in the future use SNOWPACK for near real-time assessments in which case some RCMs are not available.

L122-124: The good agreement reported by Gossart et al. (2019) is demonstrated from mean values of mean values and thus remains valid for discussing the mean climatology of Antarctica, But that reference could hardly be used to demonstrate the ability of MERRA-2 to represent climate variables at the 6-hourly time scale or at least at the characteristic time scale over which ephemeral processes such as drifting-snow compaction is active.

This is an excellent point. Because we show that SNOWPACK simulated density profiles are most sensitive to wind speed, we focus this analysis on wind speeds. A comparison of 95th percentile wind speed between 80 Antarctic AWS (Sanz Rodrigo et. al., 2012) and MERRA-2 demonstrates relatively good model performance (bias = 0.05 m s$^{-1}$, RMSE = 2.02 m s$^{-1}$) (Figure 4). We choose 95th percentiles in order to understand MERRA-2 skill in representing the strongest wind events and the corresponding strongest drifting snow events.

Although considerable differences (magnitude > 4 m s$^{-1}$) between observed and MERRA-2 simulated 95th percentile hourly wind speeds exist at individual sites in Figure 4, we see no clear systematic errors. In the revised manuscript we will include this analysis and contextualize its importance on simulated drifting snow.

L125: Could you justify why do you prefer the monthly scale when evaluation at the daily scale, or even less (SNOWPACK is forced at hourly intervals), could be similarly performed to better highlight MERRA-2 ability at reproducing the meteorology, moreover required as input in Eq. (1)? Strong wind events, during which most of drifting-snow compaction occurs, are completely smoothed out at the monthly scale.

We would argue that wind speed is the most influential and uncertain meteorological variable for quantifying high frequency variations in surface density (Figure 4). For this reason, we have focused our evaluation on 95th percentile MERRA-2 vs. observed wind speeds (please see our discussion in the previous point). As you note below, an additional high temporal evaluation of MERRA-2 surface climate is beyond the scope of this paper. Therefore we leave this remaining question for future studies. That said, we obviously need to provide SNOWPACK atmospheric forcing from some source. Since we have chosen MERRA-2, we need to understand it's underlying quality. We have chosen monthly timescales for evaluation in order to ascertain if there are any meteorological forcing biases that are so obvious as to force us to consider using another model for forcing. To clarify this point we have revised the text to the following:

"We evaluate MERRA-2 atmospheric reanalysis as forcing for SNOWPACK by comparing with monthly averaged observations at nine AWS, whose locations are shown in Fig. 2. Note that in contrast to Gossart et. al., 2019 our meteorological forcing evaluation relies on AWS located primarily in Dronning Maud Land, and therefore may not be representative of the diverse range Antarctic surface climates. By evaluating meteorological forcing at monthly timescales, we determine if there are any significant and obvious biases, however by definition we also smooth out high frequency discrepancies which may be important when evaluating instantaneous simulated density profiles."

L130: The performance of MERRA-2 at reproducing the Antarctic near-surface mete-orology (i.e., « month) is still poorly known. While this is certainly the subject of another study and lies beyond the scope of the paper, still you have all the materials required to do it, and this could be a real added value to your work while reinforcing the evaluation. This is also, again, more consistent with the time scale of drifting-snow compaction. At least could you give more statistics, i.e. RMSE and r2, which are better indicators (when combined together) than just a mean bias, to support your assertion. Moreover, I get that these 9 AWSs are not assimilated in MERRA-2 so they are all good and

independent evaluation products. But why so few AWSs when many others (>200, see Mottram et al., 2020) are available through other public sources and not necessarily assimilated in MERRA-2 (see https://gmao.gsfc.nasa.gov/pubs/docs/McCarty885.pdf)? Antarctic is vast and diverse. Most of the AWSs used here are located in DML and is a rather small sample of the climate conditions encountered across the continent. Could you add more AWSs to your analysis, or at least discuss the representativeness of these locations regarding the Antarctic climate conditions, also given that the evaluation using boreholes T and density profiles is mostly done at locations significantly away from the AWSs?

Good points all around. Given the demonstrated sensitivity to wind speeds (Figure 5) We have added a section on evaluation of 10 m wind speeds at 80 AWS scattered around Antarctica (Figure 4). We will discuss these results and their implications in detail in a new results section in the revised manuscript.

As you indicate, we believe that a detailed evaluation of MERRA-2 surface climate is beyond the scope of this study, as we primarily focus on SNOWPACK model development, and could have in fact chosen a different atmospheric model for meteorological forcing. To that point, we have added a sentence on the limitations of our MERRA-2 evaluation (see above). Furthermore we have added a sentence on limitation of only using AWSs from DML.

"Note that in contrast to Gossart et. al., 2019, our meteorological forcing evaluation relies on AWS located primarily in Dronning Maud Land, and therefore may not be representative of the diverse range Antarctic surface climates."

L133: Do you rather mean -15.1 W/2, so the applied correction correspond to the mean bias as done for ILWR? If not, where does this value come from?

A constant increase would be inappropriate when ISWR is low or zero, for example during twilight or polar night. For this reason we choose to correct MERRA-2 ISWR with a multiplicative factor which sets the average ratio of modeled and observed ISWR to 1. In our case, this multiplicative factor is 1.194 corresponding to 19.4%. Notably, as is in our case this does not guarantee that the mean bias is zero. We have tried to clarify this by updating the text to:

"We calculated an average MERRA-2 bias across all nine AWS of -15.1 W m$^{-2}$ (corresponding to 19.4 %) and -16.9 W m$^{-2}$ for ISWR and ILWR, respectively (Fig. A1). In order to reduce this bias in incoming radiation and thus better capture AWS observations, we increase MERRA-2 ISWR by 19.4 % and ILWR by 16.9 W m$^{-2}$. Note that we choose a multiplicative correction for ISWR because a constant increase would be inappropriate when ISWR is low or zero, for example during twilight or polar night."

L139: Important information are missing here, such as the initialization, the number of ice/snow layers, the vertical resolution of SNOWPACK and aggregation of new snow layers. You must elaborate on this.

We have added the following information about initialization of newly added layers and layer merging in the revised manuscript at the beginning of section 2.3.

"In our scheme, new snow layers are added on top of the modeled snow column when precipitation is present in the atmospheric forcing, in steps of 2 cm. Layers are initialized with a density given by Eq. 1 when they originate from precipitation. Layers originating from drifting snow are initialized with a density given by Eq. 4. Initial grain size for all newly added layers is 0.2 mm (Groot Zwaaftink et. al., 2013). There are two sets of microstructural properties for grain shape (dendricity and sphericity), for high and low wind speed, respectively (Groot Zwaaftink et. al., 2013). Note that precipitation is treated before assessing snow erosion, such that low density snow from precipitation can erode immediately when conditions allow. To reduce computation costs, a sophisticated snow layer merging scheme merges layers with very low ice content due to sublimation or melt and layers with similar properties (density, water content, grain size, and grain shape parameters). The criteria for layer merging are relaxed with depth, to allow for more aggressive layer merging with depth. At 10 m depth, typical layer spacing is around 10 - 20 cm. Near the surface, layers can be split to maintain a vertical resolution of a few cm near the surface, to be able to numerically represent steep temperature gradients."

L171: Defining the surface as the 1st meter is quite coarse regarding the actual thick-ness of the layer affected by surface post-depositional processes. For instance, Groot Zwaaftink et al. (2013) consider the first 10 cm. Specifying the timing at which you compare model with observations can be of significant importance for these surface layers (« 1m) depending on the recent occurrence of melt, snowfall and drifting-snow events, more importantly given that the interest of implementing a drifting-snow com-paction routine partly relies on improving representation of density at the surface at the time of drifting-snow events. I would expect more details about the comparison methodology. Do you compare observed profiles with mean modelled profiles ? for which period ?

Although, 1 m is coarse compared to some definitions, we choose to define the surface as the top 1 m in order to a) preserve consistency with the analysis presented by Alexander et. al., 2019 and b) retain a robust sample size both in terms of the number of points and their spatial distribution as most SUMup observations do not have high spatial resolution in the uppermost meter.

We agree that we left out some important information on comparison methodology. Thank you for pointing this out. We have added the following explanation: "For all models, we retrieve the simulated density profile whose time step is closest to that of the observed profile. SNOWPACK, IMAU-FDM, and GSFC-FDM report simulated density profiles every 1, 30, and 5 days respectively."

L189, "almost perfect" : Sounds a bit too emphatic. I would advise to remain neutral when describing your results. A bias value alone, even low, is not a self-consistent argument to speak of almost perfect agreement when RMSE still amounts to 2.36 ◦ C (indicating individual bias values of several degrees in some locations).

Per Reviewer 2's suggestion, we have revised "almost perfect" to "excellent".

L199: Did you follow the same spin-up procedure for your sensitivity analysis?

Good question, yes we follow the same spin-up procedure for all SNOWPACK simulations. We have clarified this in section 2.3. "In order to ensure a realistic representation of snow and firn properties throughout the entire near-surface, we complete a SNOWPACK model spinup such that simulated snow depth is 10 m at all sites before comparison with observations or other SNOWPACK simulations."

L200: Why did you choose these specific locations? Please justify and give coordinates.

We have added the following sentence: "We choose South Pole (90° S 0° E) and WAIS divide (79.5° S 112° W) because they are well known points of interest and represent distinct climactic and accumulation regimes with mean annual surface temperatures of -52.4 and -29.4 °C and accumulations of 56 and 207  kg m$^{-2}$ yr$^{-1}$ respectively."

L211-212: This is another strong argument for exploring the sensitivity of density to the derivation method for u* (z0, stability correction function) as well as for evaluating MERRA-2 wind speeds at shorter intervals.

Thank you for raising this point. We agree that there are likely uncertainties in the determination of the snow erosion, associated with the derivation of u*. These can impact density profiles. We presently use a sophisticated atmospheric stability correction method, which provides comparable results with other stability corrections (Schloegl et al., 2017). Furthermore, the by far strongest control on the amount of drifting snow is the fetch length (L, equation 3). Because the fetch length is a poorly constrained tuning parameter, we believe that an investigation of the sensitivity to roughness length and stability correction functions is not likely to produce insightful results, given that the amount of snow erosion could also be modified by the fetch length. Furthermore, we think that a meaningful study can only be performed when

including drifting snow observations. For these reasons, we would like to refrain from adding this analysis in the present study, but will consider these ideas in future research.

L214: This may be because wind maxima which control drifting-snow occurrence and thus drifting-snow compaction, are smoothed out at such a low (monthly) temporal resolution. Does this stay true if you perform a statistical evaluation of wind speed at higher temporal resolution?

Good question, we will make sure to mention this possibility in the revised manuscript. However following our analysis of 95th percentile observed wind speeds vs. MERRA-2, we still find no systematic error with regard to high wind speeds (Figure 4).

L215-216: Again, then knowing the sensitivity of your results to the derivation method of u* would be particularly interesting.

Thank you for your comment. As stated above, we have decided not to investigate the role of u* derivation method on simulated density profiles in this study.

L227-229: I couldn't agree more. This is another argument in favor of an evaluation of wind speed at higher temporal resolution. Another explanation to this result might be that drifting-snow compaction is mainly active over layers thinner than 1 m. What would the correlation become by decreasing the size of the surface layer consider here while working over shorter time scales?

For an evaluation of wind speeds, please see our analysis of 95th percentile observed wind speeds vs. MERRA-2 (Figure 4). Regarding another possible explanation, that is a very good suggestion (thanks for that!!). Indeed if we consider 1980 - 2017 maximum MERRA-2 wind speed instead of mean wind speeds, we find a significant, albeit weak, relationship between wind speed and surface density. We have noted this in the revised manuscript.

"Note that we find no significant correlation between MERRA-2 1980 - 2017 mean wind speed and observed surface density ($p = 0.14$, $R^2 = 0.03$) but do in fact find a significant, albeit weak relationship between 1980 - 2017 maximum wind speed and surface density ($p < 0.01$, $R^2 = 0.09$). Because drifting snow compaction is known to partially control snow density on daily to hourly timescales (Sommer et. al., 2018), the lack of a significant relationship between mean annual wind speed and surface snow density combined with the significant relationship between maximum wind speed and surface density indicates the importance of resolving drifting snow compaction with high temporal resolution (daily to hourly) meteorological forcing as opposed to annual means or climatology. "

L280-281: I'm wondering to which depth drift-induced compaction exerts an influence on the mean density profile, given the fact that only the surface layer receives momentum from the atmosphere and is likely affected by drifting snow. This is a very interesting question, I don't have the answer and your work is among the first to focus on this aspect. But you could give an element of response by running the model without the drift compaction routine and see how it affects density profiles (and which layers are most impacted) according to SNOWPACK pre-existing physics by comparing it with the run including drifting-snow compaction.

Excellent question. We have tried to start to answer this by running SNOWPACK with three different sets of physics in Dronning Maud Land (Figure 6). The three different sets of physics are those presented in this study (redeposit), this study without redeposition (enhanced wind compaction, Groot Zwaaftink et. al., 2013), and default alpine SNOWPACK (default). We will discuss these results in detail in a new section in the revised manuscript. But for now, we can summarize by stating that the effect of redeposition on simulated density is largest at the surface, while the difference converges with depth. This is to be expected as drift processes only operate on layers near the surface. Additionally, because at this site SNOWPACK predicts the magnitude of erosion to be approximately the same as precipitation, most snow experiences redeposition (and therefore densification) at least once. Additionally, densification is related to density, therefore layers which are less dense will compact faster, leading to the shown convergence of simulated density with depth.

L301: No new results in this section, but these are good elements of discussion. This should be entirely part of a Discussion section, or mixed with the Conclusion re-entitled Discussion and Conclusion.

You are correct that we provide no new model results in this section. We believe however that merging this with the conclusion to make a "Discussion and Conclusions" section would not improve the overall readability of the manuscript.

L317: "new snow density": specify if new=deposited (snowfall) or redeposited (drifting snow). Maybe consider staying in line with Groot Zwaaftink et al. 2013 in which "new snow" is defined as redeposited.

Thank you for your suggestion. After careful consideration, we have determined that new snow density should refer to density of new snowfall, whereas redeposited snow density should refer to the density of redeposited drifting snow. In this specific case, by "new snow density", we mean new snowfall density, rather than redeposited snow density. We have clarified this by updating to "Next, in our implementation of SNOWPACK, new snowfall density is determined by...".

L331: This section is out of the scope, of limited interest and with no scientific results. Everything here could be moved to the conclusion and resumed in one sentence informing on the availability of SNOWPACK products for other possible applications.

In the revised manuscript we have decided to limit this section to a discussion of SNOWPACK simulated surface density variability (Figure 11). Because we lack appropriate observations to validate simulated grain size and temperature profiles, we have removed these components from the section.

L350-352: You need a comparison between runs with and without the compaction routine to clearly highlight improvements and state that drifting-snow compaction is the process behind this. Besides, it would be a very interesting results that would strengthen the scientific contribution of the paper.

Agreed. We have added a section on the effect of the new SNOWPACK drifting snow compaction routine on simulated density profiles (Figure 6). In this section we show that our new scheme significantly increases simulated near-surface density, particularly in the top meter, and better matches observations.

Technical corrections

L5, "wind-driven, drifting snow": either wind-driven or drifting snow, but a combination of both is redundant.

We have modified it to "drifting snow" here and throughout the paper.

L179, "absent of very rare": Please add a reference.

Good point, this sentence deserves a reference. We have revised to "This classification yielded 122 unique observed profiles (Fig. 2) that are located primarily on the grounded ice sheet, where surface melt is limited (< 50 mm w.e. yr$^{-1}$), absent, or very rare (Trusel et. al., 2013)."

L224, "perhaps surprisingly": Avoiding subjective wording is strongly recommended.

We have removed "perhaps surprisingly".

L263: remove "at depth".

Good catch, "at depth" has been removed.

L273, "off": on?

This sentence has been revised to "To test MERRA-2 and SNOWPACK's ability to capture the SEB across a range of AIS surface climates, we compare 1980 - 2017 mean MERRA-2 surface temperature and SNOWPACK snow surface temperature with 10 m depth temperatures from 55 boreholes whose locations are show in Fig. 2."

L325: correct "comapred".

Fixed, thank you!

L336,"valide": "evaluate" is more appropriate.

Good point, we have switched to evaluate.

Agosta, C., Amory, C., Kittel, C., Orsi, A., Favier, V., Gallée, H., van den Broeke, M. R., Lenaerts, J. T. M., vanWessem, J. M., van de Berg, W. J., and Fettweis, X.: Esti- mation of the Antarctic surface mass balance using the regional climate model MAR (1979– 2015) and identification of dominant processes, The Cryosphere, 13, 281–296, https://doi.org/10.5194/tc-13-281-2019, 2019.

Amory, C., Gallée, H., Naaim-Bouvet, F., Favier, V., Vignon, E., Picard, G., Trouvilliez, A., Piard, L., Genthon, C., and Bellot, H.: Seasonal Variations in Drag Coefficient over a Sastrugi- Covered Snowfield in Coastal East Antarctica, Bound.-Lay. Meteorol., 164, 107–133, https://doi.org/10.1007/s10546-017-0242-5, 2017.

Gossart, A., Helsen, S., Lenaerts, J. T. M., Broucke, S. V., van Lipzig, N. P. M., and Souverijns, N.: An Evaluation of Surface Climatology in State-of-the- Art Reanalyses over the Antarctic Ice Sheet, Journal of Climate, 32, 6899–6915, https://doi.org/10.1175/JCLI-D-19-0030.1, 2019.

Groot Zwaaftink, C. D., Cagnati, A., Crepaz, A., Fierz, C., Macelloni, G., Valt, M., and Lehning, M.: Event-driven deposition of snow on the Antarctic Plateau: analyzing field measurements with SNOWPACK, The Cryosphere, 7, 333–347, https://www.the-cryosphere.net/7/333/2013/, 2013.

Vignon, E., Genthon, C., Barral, H., Amory, C., Picard, G., Gallée, H., Casasanta, G., and Argentini, S.: Momentum and heat-flux parametrization at Dome C, Antarctica: a sensitivity study, Bound.-Lay. Meteorol., 162, 341–367, https://doi.org/10.1007/s10546-016-0192-3, 2016.

van Wessem, J. M., van de Berg, W. J., Noël, B. P. Y., van Meijgaard, E., Amory, C., Birnbaum, G., Jakobs, C. L., KruÌLger, K., Lenaerts, J. T. M., Lhermitte, S., Ligtenberg, S. R. M., Medley, B., Reijmer, C. H., van Tricht, K., Trusel, L. D., van Ulft, L. H., Wouters, B., Wuite, J., and van den Broeke, M. R.: Modelling the climate and surface

mass balance of polar ice sheets using RACMO2 –Part 2: Antarctica (1979–2016), The Cryosphere, 12, 1479–1498, https://doi.org/10.5194/tc-12-1479-2018, 2018.

Summary

The authors apply the physically-based SNOWPACK snow model over the Antarctic ice sheet in order to simulate snow and firn densification. The SNOWPACK simulation is compared with in situ measurements and two other firn densification models. The authors find that biases in SNOWPACK are generally lower than in the two other empirically-based models, especially for locations where observations have not been used to calibrate the semi-empirical models. They suggest that in future projections of Antarctic firn densification, SNOWPACK would produce more reliable results, because of more detailed representation of physical processes, compared with simpler semi-empirical models.

General Comments

In general, the manuscript is well written and well organized. The topic is an important and relevant one, especially given the recent launch of the ICESat-2 altimetry satellite. The scheme introduced is more sophisticated than that of other models applied over the Antarctic ice sheet, and the paper therefore represents an important advance on other recent studies. The paper clearly confirms that the SNOWPACK model is capable of realistically simulating near-surface density over the Antarctic ice sheet, and has advantages over other models in that it is more detailed in its representation of physical processes governing snow evolution. I do have some concerns regarding the interpretation of results, however, as noted below. I feel that overall these revisions do not require major changes to the paper, but should be addressed before the paper can be published.

Many thanks to Reviewer 2 for their positive judgment as well as detailed and thoughtful comments. We attempt to address these comments and questions by responding individually below.

(1)  Given that the uncertainty ranges of the density simulated by the different models overlap to some degree, it is not completely clear whether there is a statistically significant difference between them, or between the models and the observations at different levels. The authors should test whether this is the case.

Good ideas! We have calculated 95% confidence intervals on the mean model biases in revised Figures 8 - 10. We will provide a detailed discussion of this analysis in the revised manuscript, but for now please see our summary.

In the case of the mean density bias for a 122 firn cores, SNOWPACK's 95% confidence interval contains zero from 0 - 7 m depth. Therefore we conclude that SNOWPACK is consistent with observations in the top 7 m. Note however that SNOWPACK and IMAU-FDM's confidence intervals overlap at 0 - 1 m and 8 - 10 m, whereas SNOWPACK and GSFC-FDM overlap at 5 - 6 m. Thus it is inappropriate to conclude that SNOWPACK produces reduced mean biases (in a statistically significant sense) throughout the entire near-surface. In the case of partitioning to high and low accumulation sites, we see a similar story, namely overlapping confidence intervals, particularly in the high accumulation category.

(2) The authors should be careful to note some of the limitations of the current implementation (e.g. the validation is over the top 10 m, not the entire firn column; and the SNOWPACK bias is larger below 6 m depth) particularly in the abstract and conclusions sections.

Agreed. In the revised manuscript we will note these limitations in both the abstract and conclusion sections.

(3) The available evidence doesn't seem to necessarily support the argument that biases are substantially larger in the semi-empirical models at locations that were not used to calibrate those models. The authors should clarify whether this is indeed the case and revise the text accordingly. It would be interesting to include both the GSFC-FDM and IMAU-FDM in this comparison if possible. Further specific comments are provided below.

We agree with the point raised by the reviewer. In the case of only examining sites not used in the GSFC-FDM calibration, we find the SNOWPACK bias magnitude decreases, whereas the GSFC-FDM bias increases. However, upon placing 95% confidence intervals on the bias means (revised Fig. 10), we see that the confidence intervals overlap. Therefore it is not statistically clear that the biases are substantially larger in the GSFC-FDM than in SNOWPACK. We will revise the text to reflect this finding.

We have included the IMAU-FDM in this analysis (Figure 10).

Specific Comments

Title: I would suggest adding SNOWPACK to the title, and mentioning the near-surface e.g. "Physics-based modeling of near-surface Antarctic snow and firn density with the SNOWPACK model". I would argue that the other models utilized here are also physically-based, they just employ simpler parameterizations for the process of firn densification.

We have updated the title to "Physics-based SNOWPACK model improves representation of near-surface Antarctic snow and firn density."

Lines 1-11: In general, some quantitative evidence should be provided here. Some of the limitations of SNOWPACK applied over Antarctica should be discussed, for example the larger bias for higher accumulation areas and the larger biases deeper in the snowpack, as well as the fact that this approach focuses on the near-surface, not the full firn column.

We agree that noting our study and findings limitations should be provided. In the revised manuscript we will incorporate these suggestions.

Lines 7-8: It would be best to quantify the magnitude of the biases here.

Agreed. Will do.

Line 9: It isn't entirely clear from this sentence that this is one of the findings of the study; it would be best to provide some quantitative results here. Also I believe the performance does degrade somewhat at these sites, just not as much as for the semi-empirical models?

We will revise this sentence to reflect the findings of our new statistical analysis in revised Figure 10.

Line 17: It would be informative to mention other methods of estimating mass balance (e.g. gravity measurements, e.g. Velicogna et al., 2020; or the input output method, e.g. Rignot et al., 2019).
Velicogna, I., Mohajerani, Y., A, G., Landerer, F., Mouginot, J., Noël, B., Rignot, E., Sutterley, T., van den Broeke, M. R., van Wessem, M., and Wiese, D. (2020) Continuity of ice sheet mass loss in Greenland and Antarctica from the GRACE and GRACE Follow-On missions. *Geophysical Research Letters* 47, e2020GL087291.
Rignot, E., Mouginot, J., Scheuchl, B., van den Broeke, M., van Wessem, M. J., and Morlighem, M. (2019) Four decades of Antarctic Ice Sheet mass balance from 1979-2017. *Proceedings of the National Academy of Sciences*, 116, 4, 1095-1103.

Good point. We didn't mean to imply that alimtetry is the only way to calculate mass balance. We have revised the manuscript by mentioning the gravity and input output methods to calculate ice sheet mass balance.

"MB is typically calculated using one of three methods, namely the input output method (Rignot et. al., 2019), gravimetry (Velicogna et. al., 2020), or satellite altimetry (e.g. Shepherd et. al., 2012; Smith et. al., 2020), the latter of which combines measurements of ice sheet volume change with modeled snow and firn density estimates."

Line 38: What is meant by "all local and temporal density variability"? No model can capture "all" variability. Please clarify.

By this we meant to communicate high frequency density variability at the snow surface. To clarify our intent, we have revised the sentence to: "However, due to their limited complexity, as measured by the inclusion of ephemeral processes such as drifting snow compaction, they cannot capture high frequency (i.e. hourly) temporal density variability at the snow surface."

Line 41: These models do employ "physical principles"; they are not entirely empirical. Suggest simply removing the phrase "rather than physical principles".

Agreed. In retrospect, the clause "rather than physical principles" is misleading and we have therefore removed it.

Lines 50-53: Describe how the model is forced, briefly.

We have updated this sentence to "In order to improve model representation of Antarctic snow and firn properties compared to semi-empirical models, we compare results from the physics-based snow model, SNOWPACK, forced by hourly weather data from MERRA-2 atmospheric reanalysis (section 2.2) to nine automatic weather stations (AWS), 55 borehole 10 m depth temperatures, and 122 observed near-surface density profiles for a total of 186 locations across the AIS."

Line 50: Instead of "we apply", do you mean "we compare results from"

We have adopted this suggestion. See previous comment for updated sentence.

Line 61: SNOWPACK also seems to include parameterizations that are empirically based. Perhaps mention explicitly how SNOWPACK is different from the other models mentioned in earlier sections.

The reviewer is absolutely right that SNOWPACK includes some empirically based parameterizations (e.g. equations 1, 3, 4), however these parameterizations have not been tuned for this specific application. Additionally, SNOWPACK differs from IMAU-FDM and GSFC-FDM in that it calculates densification via overburden stress instead of an empirical relationship. Furthermore, in contrast to semi-empirical firn densification models, SNOWPACK calculates compaction by considering snow viscosity, which is determined in part by snow microstructure (e.g. grain size, shape) internally calculated by SNOWPACK. We have clarified these points in the text by adding the following sentence.

"In order to account for this, we have implemented a new drifting snow compaction routine into the vertical, one-dimensional physics-based land-surface snow model, SNOWPACK (Bartlet and Lehning, 2002; Lehning et. al., 2002a, b), which in contrast to most existing firn models (sections 2.5 - 2.6), calculates densification using an overburden stress formulation as opposed to an empirical relationship and explicitly determines snow viscosity by calculating the snow microstructure (e.g. grain size and shape) and temperature."

Line 75: Perhaps change "new drifting snow compaction routine" to "new snow compaction routine", as drifting snow is just a component of the routine.

Our model does indeed rely on existing modules, e.g. overburden stress, new snow density, snow microstructure, etc. However, since the focus of this study is on the treatment of drifting snow we have decided to keep the term "new drifting snow compaction routine" in order to emphasize our new contribution to SNOWPACK.

Line 80: Can the authors briefly note how this parameterization is derived?

We have added the following note on how the parameterization was developed "which is a multiple linear regression derived from observations in the Swiss Alps (Lehning et. al., 2002)".

Lines 87-88: How much do these parameters change the comparison with observed profiles. Provide some additional details either in the main manuscript or a supplemental section.

We have added the following information: "However, we find that the parameter tuning proposed by Steger et. al., (2017) leads to significantly overestimated densities (bias > 50 kg/m^3) in the dry snow zone of Antarctica. Therefore we revert to original SNOWPACK viscosity parameters by setting $Q_s$ and $\beta$ to 67,000 J mol$^{-1}$ and 0.7, respectively."

Line 90: Briefly explain the physical meaning of the "threshold friction velocity".

Great idea - we have added the following definition, "the minimum friction velocity at which surface snow grains are mobilized by the wind".

Line 124: Is this a bias over the entire Antarctic ice sheet? Are there spatial variations in the bias?

To answer these questions we have revised the text to the following:

"however MERRA-2 appears to overestimate SMB, with an ice sheet wide mean absolute error of 58.5 kg m$^{-2}$ yr$^{-1}$. According to Gossart et. al., 2019, MERRA-2 well captures coastal and ice shelf SMB but generally overestimates SMB in the escarpment zone and at elevations from 500 - 3000 m. "

Lines 130-131: Is there a reference for these statements?

Thank you for bringing this to our attention. This sentence is actually in reference to our own analysis. But admittingly this was not clear. We have therefore revised the sentence to "According to our analysis and consistent with the findings of Lenaerts et. al., 2017 and Gossart et. al., 2019, MERRA-2  well captures observed 2 m air temperature, relative humidity, and wind speed, but significantly underestimates both ISWR and ILWR.".

Line 133: Why use 19.4% and not 15.1 W m-2 ?

A constant increase would be inappropriate when ISWR is low or zero, for example during twilight or polar night. For this reason we choose to correct MERRA-2 ISWR with a multiplicative factor which sets the average ratio of modeled and observed ISWR to 1. In our case, this multiplicative factor is 1.194 corresponding to 19.4%. Notably, as is in our case this does not guarantee that the mean bias is zero. We have tried to clarify this by updating the text to:

"We calculated an average MERRA-2 bias across all nine AWS of -15.1 W m$^{-2}$(corresponding to 19.4 %) and -16.9  W m$^{-2}$ for ISWR and ILWR, respectively (Fig. A1). In order to reduce this bias in incoming radiation and thus better capture AWS observations, we increase MERRA-2 ISWR by 19.4 % and ILWR by 16.9  W m$^{-2}$. Note that we choose a multiplicative correction for ISWR because a constant increase would be inappropriate when ISWR is low or zero, for example during twilight or polar night."

Line 134: Why is there still a bias after the bias is removed?

We attribute the remaining bias to the fact that SNOWPACK uses an internal meteorological data preprocessing library which contains a variety of filters. Therefore modifications to initial meteorological data (i.e. increasing MERRA-2 ILWR by 16.9  W m$^{-2}$) do not map onto SNOWPACK forcing in a one to one way. Admittingly this is not ideal, however this remaining bias (0.6  W m$^{-2}$) is small both compared to MERRA-2 and observational uncertainties.

Lines 154-170: It would be helpful here to describe these two models in a bit more detail, in particular to highlight how they differ from SNOWPACK in terms of key

physical processes (e.g. compaction), as the model differences are important to the conclusions of the study.

Good point, this is important to clarify. Therefore we have added the following text for IMAU-FDM to section 2.5:

"In contrast to SNOWPACK, which relies on an overburden stress compaction scheme, IMAU-FDM uses a calibrated semi-empirical dry snow densification scheme based on Arthern et. al., 2010, which describes densification as a function of density as well as annual average accumulation and temperature. In further contrast to SNOWPACK, the IMAU-FDM parameterizes new snow density as a function of annual average, rather than hourly, meteorology and currently includes no post deposition mechanism to increase surface snow density due to drifting snow processes."

For GSFC-FDM we have revised to the following in section 2.6:

"The dry snow and firn compaction model, based on Arthern et. al., 2010, was calibrated to observed depth-density profiles from both Greenland and Antarctica. A simple initial density scheme was implemented based on mean annual MERRA-2 climate, which provides a spatially variable initial density that does not, in contrast to SNOWPACK, vary in time or vary due to drifting snow processes."

Line 166: Explain the meaning of "replay".

In retrospect, "replay" is a highly-specific and poorly defined term that we believe is beyond the scope of this study to explain in detail. Therefore we replace "replay" with "reanalysis" and point the reader to the appropriate reference for more precise details.

Line 194: Suggest changing "reduction in both RMSE..." to "statistically significant reduction in both RMSE..."

We have implemented this suggestion.

Line 196: This section could potentially be moved to later in the manuscript.It might logically follow the section on comparison with observations. 23.

Thank you for your suggestion. We have thoroughly considered it, but ultimately decided to keep this section where it is in order to contextualise SNOWPACK biases within the context of uncertainties in atmospheric forcing.

Line 200: Clarify why these two stations were chosen.

We have added the following sentence: "We choose South Pole (90° S 0° E) and WAIS divide (79.5° S 112° W) because they are well known points of interest and represent distinct climactic and accumulation regimes with mean annual surface temperatures of -52.4 and -29.4 °C and accumulations of 56 and 207 kg m$^{-2}$ yr$^{-1}$ respectively."

Line 221: It is a bit unclear what is meant by "we tested for explanatory variables". Please clarify.

We have revised the sentence to "Since it is known that meteorological conditions including annual accumulation and temperature influence Antarctic snow and firn density (Herron and Langway, 1980), we tested this hypothesis."

Lines 245-247: It might be useful to have a table here for the bias and RMSE for different models above and below 400 kg m-3.

Thank you for your suggestion. By including this sentence, we intend to demonstrate that all models see degradation when observed surface density exceeds 400 kg m$^{-3}$. For this reason we have added the following table to the revised manuscript, which includes surface density bias and RMSE for all three models at all observations and at observations whose surface density exceeds 400 kg m$^{-3}$.

| | SNOWPACK | GSFC-FDM | IMAU-FDM |
|---|---|---|---|
| Bias (kg m$^{-3}$) at all sites | -8.2 | 20.4 | -20.4 |
| Bias (kg m$^{-3}$) at high surface density (> 400 kg m$^{-3}$) sites | -23.7 | -20.4 | -65.4 |
| RMSE (kg m$^{-3}$) at all sites | 45.3 | 38.5 | 40.7 |
| RMSE (kg m$^{-3}$) at high surface density (> 400 kg m$^{-3}$) sites | 65.8 | 35.6 | 74.7 |

Line 253: This sentence is confusing. Suggest revising to read something like: "Additionally, we cannot rule out the possibility of larger errors in the observational data for densities above 400 kg m-3."

Thank you for your suggestion. We have adopted this.

Line 256: The SUMup dataset does include information on measurement methods. It might be interesting to see if dividing by measurement method changes these biases in any way.

Thank you for your suggestion. Indeed, SUMup does contain information about measurement methods and it would be interesting to understand how the measurement method affects the bias. However, because this study revolves primarily around model development, we deem the proposed analysis to be outside the scope of this study. With that in mind, we have included your suggestion as a point of future analysis.

Lines 259-271, Fig. 6: Can the authors note whether the differences are statistically significant? It might also be useful to provide an uncertainty range on the biases. Also, at first glance at it appears that all the model simulation uncertainty ranges overlap in Fig. 6, but this is not the case. Perhaps the figure can be modified slightly to make this clearer, e.g. changing the transparency for different models or changing the colors. (Not sure how easy this would be).

We have adjusted the transparency of shading in revised figures 8 - 10. We think this improves readability. With regard to statistical significance, please see our comment on point (1) at the top of our response to your review.

Lines 293-294: It seems this would not be difficult to find out? It would also be interesting to see the IMAU-FDM results.

Fortunately we were able to retrieve information on which firn cores were used to calibrate the IMAU-FDM! Note that for IMAU-FDM, the mean bias goes from -32.5 $\text{kg m}^{-3}$ at all sites to -20.4 $\text{kg m}^{-3}$ at only the 69 independent sites (those not used in calibration). Thus, in contrast to the GSFC-FDM, IMAU-FDM actually sees a reduction in bias mean magnitude. However, this bias is still greater in magnitude to that of SNOWPACK (-9.7 $\text{kg m}^{-3}$ and 1.1 $\text{kg m}^{-3}$ for all sites and the 69 independent sites, respectively).

Lines 296-297: From Fig. 8, it actually looks like there is a larger change in the SNOWPACK density bias (at least at different levels). The numbers here do not seem to match with the figure. Please clarify.

Thank you for pointing out this confusion. In the original paper we report the mean bias magnitude. To simplify the interpretation, we have decided to now report the mean bias with depth.

The SNOWPACK mean bias went from -9.7 to 1.1 kg m$^{-3}$. Whereas the GSFC-FDM mean bias went from 15.5 to 22.3 kg m$^{-3}$. Therefore the SNOWPACK bias did change (in an absolute sense) by more than the GSFC-FDM bias. However, the SNOWPACK bias magnitude is in fact reduced, whereas the GSFC-FDM bias magnitude increases. We have modified the corresponding sentences in the paper to the following:

"By examining only the 69 independent sites, the mean GSFC-FDM density bias increases throughout the near-surface compared to all 122 sites, from 15.5 to 22.3 kg m$^{-3}$ whereas the SNOWPACK mean density bias goes from -9.7 to 1.1kg m$^{-3}$ (Fig. 10). Because the mean near-surface density bias magnitude increases for GSFC-FDM while decreasing for SNOWPACK, we conclude that the semi-empirical firn densification model may exhibit reduced model performance at sites not included in density calibration."

Lines 335-336: This portion is interesting but seems disconnected from the rest of the manuscript. Perhaps these temporal variations could be placed in the context of temporal variations from in situ data. Are there any locations where a timeseries of measurements is available that could be compared with the SNOWPACK runs?

Unfortunately the SUMup density data set used in this study, does not have any high frequency (sub-annual to hourly) density observations that could be used for the suggested analysis. That said, the suggested analysis is likewise interesting to us and also relevant for interpreting our new compaction scheme.

Line 342: Without validation of the temporal variability of the in situ measurements, I'm not sure the model results would qualify as "evidence". Please revise.

We agree that because SNOWPACK surface density temporal variability is not evaluated against observations, it should be considered suggestive of, rather than evidence for significant surface density variability. We have tried to clarify this by revising to the following:

"By describing snow cover properties at South Pole (mean annual $T_s$ = -52.4 °C and SMB = 56 kg m$^{-2}$ yr$^{-1}$) as well as WAIS (mean annual $T_s$ = -29.4 °C and SMB = 207 kg m$^{-2}$ yr$^{-1}$), we present model results which suggest significant regional variability in near-surface temperature as well as both regional and temporal variability in surface density and grain diameter."

Lines 360-364: I'm not sure these statement is completely supported by the results. For example, SNOWPACK seems to show a larger bias at higher accumulation locations, and the SNOWPACK and the GSFC-FDM both seem to show a positive bias in locations that were not used to constrain GSFC-FDM between 0 and 6 m in depth. In

general, however, I would agree that including a more physically realistic simulation of snowpack processes should produce a better projection of future conditions. Perhaps revise this statement to note that this is likely the case, but not entirely certain.

We have rephrased this section to use more conservative language by revising to "For this reason, SNOWPACK, when compared to GSFC-FDM, may perhaps simulate snow and firn density more accurately at sites whose density observations are not included in the GSFC-FDM density calibration. Because SNOWPACK outperforms the GSFC-FDM at sites not included in calibration, SNOWPACK, compared to semi-empirical models, could possibly simulate firn density more accurately in regions without extensive observations or under future climate scenarios, where firn properties are expected to diverge from their current state."

Technical Corrections

Line 14: Change "with an increasing..." to "at an increasing..."

Done.

Line 91: Change "from MERRA-2" to "from the MERRA-2".

Fixed, thank you.

Fig. 2 caption: Change "SNOWPACK simulations" to "SNOWPACK simulation locations" for clarity. Note that the borehole depths are 10 m below the surface for clarity.

Thank you for these suggestions. The Figure 2 caption now reads "Map of SNOWPACK simulation locations over the Antarctic ice sheet. SNOWPACK simulation locations at 122 observed density profiles (upside down blue triangles), 55 borehole 10 m depth temperature measurements (red triangles), and nine automatic weather stations (AWS, yellow circles) plotted over MERRA-2 1980 - 2017 mean annual SMB.."

Line 172: Change "as depths" to "the average density between depths of"

Good suggestion, we have implemented this.

Line 185: Change "of average" to "of the average".

Fixed!

Line 189: Suggest changing "almost perfect" to "an excellent".

Thank you for your suggestion, we have implemented this change.

Line 198: Change "is bias-corrected MERRA-2" to "in bias-corrected MERRA-2".

Fixed, thank you.

Line 215: Change "windspeed represents" to "density variations due to uncertainty in windspeed represent".

Good suggestion! We have updated this sentence to "We must therefore acknowledge that density variations due to uncertainty in wind speed represent the largest source of uncertainty with regard to SNOWPACK simulated near-surface density, and in fact, exceeds uncertainties arising from firn densification model choice."

Line 219: Change "observations" to "observed density values".

We have updated "Observations" to "Observed densities".

Lines 259-260: This sentence is quite wordy. Suggest revising, e.g. "In a comparison at 122 observed density profiles, SNOWPACK exhibits a lower bias compared to IMAU-FDM for the entire near-surface, and a lower bias compared to GSFC FDM between from 0 to 7 m depth (Fig. 6)".

Good point, we have implemented your suggestion.

Lines 273-274: Suggest changing to read "low SMB categories by applying a threshold of 200 kg m-2 yr-1 to MERRA-2 mean annual SMB (Fig. 7)."

Thank you for your suggestion. We have reformulated to "we partition the 122 observed density profiles into 35 high and 87 low SMB categories by applying a mean annual SMB threshold of 200 kg $m^{-2}$ $yr^{-1}$ according to MERRA-2 (Fig. 9)."

Line 276: Change "reduced" to "lower" or "smaller"

Changed to lower.

Line 277: Again change "reduced" to "lower" or "smaller".

Changed to smaller.

Caption, Fig. 7: Change "GSFM-FDM" to "GSFC-FDM". Change "a MERRA-2 1980-2017 mean annual SMB threshold of 200 kg m-2 yr-1" to "a 200 kg m-2 yr-1 threshold applied to MERRA-2 1980-2017 mean annual SMB."

Thanks for catching the typo! We have fixed this as well as implementing your suggestions.

Lines 303-304: Change "as well as their different level of process representation complexity" to "as well as their different level of complexity in representing physical processes."

Good suggestion! We will use this.

Line 325: Change "compared" to "compared".

Fixed!

Line 373: The heading for Appendix A is out of place.

Fixed, thank you for pointing this out.

**Figures**

[Figure]

Figure 1: Schematic of SNOWPACK drifting snow compaction. When the friction velocity exceeds the snow threshold friction velocity ($u_* > u_{*th}$), initial lower density surface snow (left) is eroded by the wind, suspended above the snow surface (middle), and then redeposited with a higher density (right).

[Figure]

Figure 2: Map of SNOWPACK simulation locations over the Antarctic ice sheet. SNOWPACK simulation locations at 122 observed density profiles (upside down blue triangles), 55 borehole 10 m depth temperature measurements (red triangles), and nine automatic weather stations (AWS, yellow circles) plotted over MERRA-2 1980 - 2017 mean annual SMB.

[Figure]

Figure 3: Modeled surface temperature evaluation. Comparison between observed borehole 10 m depth temperature (°C) and 1980 - 2017 model average non-bias corrected SNOWPACK (blue), bias corrected SNOWPACK (black), and MERRA-2 (red) surface temperature $T_s$ (°C). The dashed black line represents a one to one line. Solid lines represent linear regressions. R-squared values, mean bias, and RMSE are reported for non-bias corrected SNOWPACK (blue text), bias corrected SNOWPACK (black text), and MERRA-2 (red text).

[Figure]

Figure 4: Evaluation of MERRA-2 95th percentile hourly wind speeds. Scatter plot (left) of observed 95th percentile hourly wind speeds vs. MERRA-2 simulated 95th percentile hourly wind speeds at 80 automatic weather stations scattered across the Antarctic ice sheet (right).

[Figure]

Figure 5: SNOWPACK simulated density profile sensitivity to uncertainties in atmospheric forcing. SNOWPACK simulated density profile sensitivity at South Pole (a,b,c) and WAIS (d,e,f), to wind speed (a,d), temperature (b,e), and precipitation (c,f). The increased wind speed (+ 2.4 m s⁻¹), increased temperature (+ 3.1 °C), and reduced precipitation (- 20 %) perturbations are shown in red, while decreased wind speed (- 2.4 m s⁻¹), decreased temperature (- 3.1 °C), and increased precipitation (+ 20 %) perturbations are shown in blue. In all panels, density profiles are valid for December 31st, 2017 and black curves represent unperturbed atmospheric forcing simulations.

[Figure]

Figure 6: SNOWPACK simulated density profiles sensitivity to prescribed snow physics. Observed (blue), SNOWPACK simulated redeposit (black), default (red), and enhanced wind compaction (magenta) near-surface density profiles in Dronning Maud Land (75.0 °S, 0.0 °W, 2891m).

[Figure]

Figure 7: Modeled and observed surface density comparison. Scatter plot of observed vs. SNOWPACK modeled surface density at 79 locations across the AIS (a). Scatter plot of observed vs. GSFC-FDM modeled surface density at 79 locations across the AIS (c). Scatter plot of observed vs. IMAU-FDM modeled surface density at 79 locations across the AIS (E). Map of SNOWPACK minus observed surface density at 79 locations across the AIS (b). Map of GSFC-FDM minus observed surface density at 79 locations across the AIS (d). Map of IMAU-FDM minus observed surface density at 79 locations across the AIS (f). In panels a, c, and e, dashed black lines represent one to one lines while solid lines represent linear regressions.

[Figure]

Figure 8: Density profile comparison. Mean observed (blue dashed), SNOWPACK modeled (black dashed), GSFC-FDM (red dashed) and IMAU-FDM modeled (yellow dashed) near surface (0 - 10 m) density profiles at 122 locations across the AIS (a). Shading represents plus and minus one standard deviation across observed and modeled density profiles. Mean SNOWPACK minus observed (black dashed), GSFC-FDM minus observed (red dashed), and IMAU-FDM minus observed (yellow dashed) density profiles at 122 locations across the AIS (b). Error bars represent 95 % confidence intervals on bias means.

[Figure]

Figure 9: High and low SMB density profile comparison. Average observed (blue dashed), SNOWPACK modeled (black dashed), GSFC-FDM modeled (red dashed), and IMAU-FDM modeled (yellow dashed) near surface (0 - 10 m) density profiles at high (a) and low (d) SMB sites. Shading represents plus and minus one standard deviation across observed and modeled density profiles. Mean SNOWPACK minus observed (black dashed), GSFC-FDM minus observed (red dashed), and IMAU-FDM minus observed (yellow dashed) density profiles at high (b) and low (e) SMB sites. Error bars represent 95 % confidence intervals on bias means. Maps of locations of high (c) and low (f) SMB sites. High and low SMB sites are delineated by a 200 kg m$^{-3}$ yr$^{-1}$ threshold applied to MERRA-2 1980 - 2017 mean annual SMB.

[Figure]

Figure 10: SNOWPACK, IMAU-FDM, and GSFC-FDM density simulations at sites not included in the semi-empirical models' calibration. Average observed (blue), SNOWPACK (black), IMAU-FDM (yellow), and GSFC-FDM modeled (red) near surface (0 - 10 m) density profiles at 69 sites not included in the IMAU-FDM and GSFC-FDM density calibrations (a). Shading represents plus and minus one standard deviation across 69 observed and modeled density profiles. Mean SNOWPACK minus observed (black dashed), IMAU-FDM minus observed (yellow), and GSFC-FDM minus observed (red dashed) density profiles at the 69 sites not included in the IMAU-FDM and GSFC-FDM density calibrations (b). Error bars represent 95 % confidence intervals on bias means. Map of 69 observed density profiles not included in the IMAU-FDM and GSFC-FDM density calibrations (c).

[Figure]

Figure 11: SNOWPACK simulated surface snow density. Map of 1980 - 2017 mean SNOWPACK surface density (kg m$^{-3}$, a). Map of SNOWPACK simulated surface density variability, defined as the 1980 - 2017 maximum difference between summer and winter mean surface density (kg m$^{-3}$}, b). Time series of 2010 - 2015 SNOWPACK simulated average surface density (kg m$^{-3}$) at South Pole (blue) and WAIS (red, c).

[Figure]

Figure A1: MERRA-2 incoming shortwave and longwave radiation bias correction. Comparison between monthly averaged AWS observed and MERRA-2 modeled incoming shortwave radiation (a) and longwave radiation (b) at nine automatic weather stations. Non-bias corrected MERRA-2 radiative fluxes are shown in blue while bias corrected fluxes are shown in red. Mean MERRA-2 root mean square error (RMSE, W m$^{-2}$) and bias (W m$^{-2}$) are shown in blue and red text for non bias corrected and bias corrected radiative fluxes, respectively. Dashed black line represents a one to one line.

**References**

Alexander, P. M., M. Tedesco, L. Koenig, and X. Fettweis. "Evaluating a Regional Climate Model Simulation of Greenland Ice Sheet Snow and Firn Density for Improved Surface Mass Balance Estimates." *Geophysical Research Letters* 46, no. 21 (November 16, 2019): 12073–82. https://doi.org/10.1029/2019GL084101.

Bintanja, Richard. "The Interaction between Drifting Snow and Atmospheric Turbulence." *Annals of Glaciology* 26 (1998): 167–73. https://doi.org/10.3189/1998AoG26-1-167-173.

Amory, Charles, Hubert Gallée, Florence Naaim-Bouvet, Vincent Favier, Etienne Vignon, Ghislain Picard, Alexandre Trouvilliez, Luc Piard, Christophe Genthon, and Hervé Bellot. "Seasonal Variations in Drag Coefficient over a Sastrugi-Covered Snowfield in Coastal East Antarctica." *Boundary-Layer Meteorology* 164, no. 1 (July 2017): 107–33. https://doi.org/10.1007/s10546-017-0242-5.

Gossart, A., S. Helsen, J. T. M. Lenaerts, S. Vanden Broucke, N. P. M. van Lipzig, and N. Souverijns. "An Evaluation of Surface Climatology in State-of-the-Art Reanalyses over the Antarctic Ice Sheet." *Journal of Climate* 32, no. 20 (October 2019): 6899–6915. https://doi.org/10.1175/JCLI-D-19-0030.1.

Groot Zwaaftink, C. D., A. Cagnati, A. Crepaz, C. Fierz, G. Macelloni, M. Valt, and M. Lehning. "Event-Driven Deposition of Snow on the Antarctic Plateau: Analyzing Field Measurements with SNOWPACK." *The Cryosphere* 7, no. 1 (February 27, 2013): 333–47. https://doi.org/10.5194/tc-7-333-2013.

Lehning, Michael, Perry Bartelt, Bob Brown, and Charles Fierz. "A Physical SNOWPACK Model for the Swiss Avalanche Warning Part III: Meteorological Forcing, Thin Layer Formation and Evaluation." *Cold Regions Science and Technology*, 2002, 16.

Lenaerts, Jan T. M., Kristof Van Tricht, Stef Lhermitte, and Tristan S. L'Ecuyer. "Polar Clouds and Radiation in Satellite Observations, Reanalyses, and Climate Models: POLAR CLOUDS AND RADIATION." *Geophysical Research Letters* 44, no. 7 (April 16, 2017): 3355–64. https://doi.org/10.1002/2016GL072242.

Michlmayr, Gernot, Michael Lehning, Gernot Koboltschnig, Hubert Holzmann, Massimiliano Zappa, Rebecca Mott, and Wolfgang Schöner. "Application of the Alpine 3D Model for Glacier Mass Balance and Glacier Runoff Studies at Goldbergkees,

Austria." *Hydrological Processes* 22, no. 19 (September 15, 2008): 3941–49. https://doi.org/10.1002/hyp.7102.

Rignot, Eric, Jérémie Mouginot, Bernd Scheuchl, Michiel van den Broeke, Melchior J. van Wessem, and Mathieu Morlighem. "Four Decades of Antarctic Ice Sheet Mass Balance from 1979–2017." *Proceedings of the National Academy of Sciences* 116, no. 4 (January 22, 2019): 1095–1103. https://doi.org/10.1073/pnas.1812883116.

Sanz Rodrigo, Javier, Jean-Marie Buchlin, Jeroen van Beeck, Jan T. M. Lenaerts, and Michiel R. van den Broeke. "Evaluation of the Antarctic Surface Wind Climate from ERA Reanalyses and RACMO2/ANT Simulations Based on Automatic Weather Stations." *Climate Dynamics* 40, no. 1–2 (January 2013): 353–76. https://doi.org/10.1007/s00382-012-1396-y.

Schlögl, Sebastian, Michael Lehning, Kouichi Nishimura, Hendrik Huwald, Nicolas J. Cullen, and Rebecca Mott. "How Do Stability Corrections Perform in the Stable Boundary Layer Over Snow?" *Boundary-Layer Meteorology* 165, no. 1 (October 2017): 161–80. https://doi.org/10.1007/s10546-017-0262-1.

Smeets, C. J. P. P., and M. R. van den Broeke. "Temporal and Spatial Variations of the Aerodynamic Roughness Length in the Ablation Zone of the Greenland Ice Sheet." *Boundary-Layer Meteorology* 128, no. 3 (September 2008): 315–38. https://doi.org/10.1007/s10546-008-9291-0.

Sommer, Christian Gabriel, Nander Wever, Charles Fierz, and Michael Lehning. "Investigation of a Wind-Packing Event in Queen Maud Land, Antarctica." *The Cryosphere* 12, no. 9 (September 14, 2018): 2923–39. https://doi.org/10.5194/tc-12-2923-2018.

Trusel, Luke D., Karen E. Frey, Sarah B. Das, Peter Kuipers Munneke, and Michiel R. van den Broeke. "Satellite-Based Estimates of Antarctic Surface Meltwater Fluxes: SATELLITE-BASED ANTARCTIC MELT FLUXES." *Geophysical Research Letters* 40, no. 23 (December 16, 2013): 6148–53. https://doi.org/10.1002/2013GL058138.

Velicogna, Isabella, Yara Mohajerani, Geruo A, Felix Landerer, Jeremie Mouginot, Brice Noel, Eric Rignot, et al. "Continuity of Ice Sheet Mass Loss in Greenland and Antarctica From the GRACE and GRACE Follow-On Missions." *Geophysical Research Letters* 47, no. 8 (April 28, 2020). https://doi.org/10.1029/2020GL087291.

Vignon, E., Genthon, C., Barral, H., Amory, C., Picard, G., Gallée, H., Casasanta, G., and Argentini, S.: Momentum and heat-flux parametrization at Dome C, Antarctica: a

sensitivity study, Bound.-Lay. Meteorol., 162, 341–367, https://doi.org/10.1007/s10546-016-0192-3, 2016.

---

## Referee Report (RR1)

Many thanks to the authors for their detailed responses. The revised version has improved significantly and they have responded to my comments quite well. I think the paper now warrants publication in its current form, providing that the authors take into account the following minor suggestions:

P2L35 (revised version): Saltation is physically defined as the motions of particle within the first 10 centimeters above ground (e.g. Pomeroy, 1989), not 2 m. I see a less major issue at referring to drifting snow only as saltating snow as long as it is explicitly mentioned in the text, although I'm not aware of any reference to rely on for such a statement. But surely saltation could not be reasonably defined as the motions of particles from 0 to 2 m. Please correct. The caption of Fig. 1 could also be adapted ("mobilized" or "put in saltation" instead of suspended?) as the model more likely represents the effect of saltation rather the a full saltation+suspension layer, as explained in Section 2.1.

P5L112: I'd like to see this value for roughness length discussed and put a bit in perspective of the existing observed values over Antarctica (see for instance Amory et al. 2017 for a review but plenty other references are possible).

Amory, C., Gallée, H., Naaim-Bouvet, F., Favier, V., Vignon, E., Picard, G., Trouvilliez, A., Piard, L., Genthon, C., and Bellot, H.: Seasonal Variations in Drag Coefficient over a Sastrugi-Covered Snowfield in Coastal East Antarctica, Bound.-Lay. Meteorol., 164, 107–133, https://doi.org/10.1007/s10546-017-0242-5, 2017.

Pomeroy, J. W., A process-based model of snow drifting, Ann. Glaciol., 13,237-240, 1989.

---

## Author Response (AR2)

Thank you to editor Xavier Fettweis and reviewer Charles Amory for their continued reviews and feedback. We appreciate both of you taking the time to improve our manuscript. In order to address your comments, we have responded individually in blue. On behalf of all coauthors, we once again thank you for your time and effort.

On behalf all authors,

Eric Keenan

**Editor Review - Xavier Fettweis**
Comments to the Author:

Dear Authors,

I'm happy to accept your paper for publication in TC.

Thank you for accepting our manuscript! We have responded to your comments individually below.

Some minor changes ("reviewed" by me only) are nevertheless needed before final acceptation. In addition to the minor changes requested by the 1st reviewer (thanks to him!), could you list the statistics (mean bias + RMSE) of the comparison with the 69 independent sites (lines 420-432) in Table 1? The impact of using 69 vs 122 sites on the comparison will be more clear…

Thank you for the suggestion. Because the statistics you are referring to reflect the entire top 10 m, rather than the top 1 m (as in table 1), we have decided to add a new table (table 2, copied below).

|  | SNOWPACK | GSFC-FDM | IMAU-FDM |
|---|---|---|---|
| Bias (kg m$^{-3}$) at all sites | -9.7 | 15.5 | -32.5 |
| Bias (kg m$^{-3}$) at 69 independent sites | 1.1 | 22.3 | -20.4 |
| RMSE (kg m$^{-3}$) at all sites | 48.3 | 36.8 | 51.5 |
| RMSE (kg m$^{-3}$) at at 69 independent sites | 47.6 | 41.6 | 46.1 |

Thanks and best regards,
Xavier F.

PS: In Fig 11: the bleu of the time series is not the same bleu than the one used for the legend (South Pole). Idem in Fig3 with the listed statistics.

Good catch, thanks for noticing! We have fixed this issue in figures 3, 11, and A1.

**Review 1 - Charles Amory**

Many thanks to the authors for their detailed responses. The revised version has improved significantly and they have responded to my comments quite well. I think the paper now warrants publication in its current form, providing that the authors take into account the following minor suggestions:

Thank you for the positive comments. We greatly appreciate you taking the time to review our manuscript a second time. We have responded to your comments individually below.

P2L35 (revised version): Saltation is physically defined as the motions of particle within the first 10 centimeters above ground (e.g. Pomeroy, 1989), not 2 m. I see a less major issue at referring to drifting snow only as saltating snow as long as it is explicitly mentioned in the text, although I'm not aware of any reference to rely on for such a statement. But surely saltation could not be reasonably defined as the motions of particles from 0 to 2 m. Please correct. The caption of Fig. 1 could also be adapted ("mobilized" or "put in saltation" instead of suspended?) as the model more likely represents the effect of saltation rather the a full saltation+suspension layer, as explained in Section 2.1.

Good point. Upon further investigation, we agree that defining the saltation layer as the lowermost 2 m is not appropriate. However, wind-mobilized particles will be deposited only via the saltation layer, which indeed is generally considered to be the lowermost 10 cm. To address this we have modified the sentence to the following:

L34: "In particular, surface snow and firn density are known to be strongly impacted by wind-driven compaction, a process hereafter referred to as drifting snow compaction, whereby mobilized snow particles in the saltation layer, defined as the lowermost 10 cm of the atmosphere (Pomeroy 1989), break apart upon collision with the snow surface."

In the Figure 1 caption we have replaced "suspended" with "mobilized".

P5L112: I'd like to see this value for roughness length discussed and put a bit in perspective of the existing observed values over Antarctica (see for instance Amory et al. 2017 for a review but plenty other references are possible).

Good idea, we have noted that a roughness length of 2 mm lies on the high end of observed values.

L113: "Note that although 2 mm is approximately an order of magnitude larger than typically observed values over the AIS (e.g. Vignot et al., 2017), even larger values have been observed in sastrugi dominated environments (Amory et al., 2017) where drifting snow erosion and deposition is common."

Amory, C., Gallée, H., Naaim-Bouvet, F., Favier, V., Vignon, E., Picard, G., Trouvilliez, A., Piard, L., Genthon, C., and Bellot, H.: Seasonal Variations in Drag Coefficient over a

Sastrugi- Covered Snowfield in Coastal East Antarctica, Bound.-Lay. Meteorol., 164, 107–133, https://doi.org/10.1007/s10546-017-0242-5, 2017.

Pomeroy, J. W., A process-based model of snow drifting, Ann. Glaciol., 13,237-240, 1989.

**References**

Amory, C., Gallée, H., Naaim-Bouvet, F., Favier, V., Vignon, E., Picard, G., Trouvilliez, A., Piard, L., Genthon, C., and Bellot, H.: Seasonal Variations in Drag Coefficient over a Sastrugi- Covered Snowfield in Coastal East Antarctica, Bound.-Lay. Meteorol., 164, 107–133, https://doi.org/10.1007/s10546-017-0242-5, 2017.

Pomeroy, J. W., A process-based model of snow drifting, Ann. Glaciol., 13,237-240, 1989.

Vignon, Etienne, Christophe Genthon, Hélène Barral, Charles Amory, Ghislain Picard, Hubert Gallée, Giampietro Casasanta, and Stefania Argentini. "Momentum- and Heat-Flux Parametrization at Dome C, Antarctica: A Sensitivity Study." *Boundary-Layer Meteorology* 162, no. 2 (February 2017): 341–67. https://doi.org/10.1007/s10546-016-0192-3.